**1** **Brief communication**

**2** **Nonlinear sensitivity of glacier-mass balance to climate attested by**

**3** **temperature-index models**

**4** *Christian Vincent and Emmanuel Thibert*

**6** *Université Grenoble Alpes, CNRS, IRD, Grenoble-INP, INRAE, Institut des Géosciences de*

**7** *l'Environnement (IGE, UMR 5001), F-38000 Grenoble, France.*

**9** Correspondence : *Christian Vincent ([christian.vincent@univ-grenoble-alpes.fr](mailto:christian.vincent@univ-grenoble-alpes.fr)) and Emmanuel Thibert*

**10** *(emmanuel.thibert@inrae.fr)*

**12** **Abstract**

**13** Temperature-index models have been widely used for glacier-mass projections spanning the $21^{st}$

**14** century. The ability of temperature-index models to capture nonlinear responses of glacier surface-mass

**15** balance (SMB) to high deviations in air temperature and solid precipitation has recently been ~~questioned~~

**16** discovered in the context of~~by~~ mass-balance simulations employing advanced machine-learning

**17** techniques. Here, we performed numerical experiments with a classic ~~and simple~~ temperature-index

**18** model and confirmed that such models are capable of detecting nonlinear responses of glacier SMB to

**19** temperature and precipitation changes. Nonlinearities derive from the change of the degree-day factor

**20** over the ablation season and from the lengthening of the ablation season.

**21**

**22** **Introduction**

**23** Glacier SMB projections in response to climate change up to the end of the $21^{st}$ century can be analysed

**24** via physical approaches using energy-balance calculations and empirical approaches linking simple

**25** meteorological variables to SMB such as temperature-index models. Most glacier-mass projections in

**26** response to climate change in large-scale studies spanning the $21^{st}$ century have been based on

temperature-index models (Huss and Hock, 2015; Fox-Kemper *et al*., 2021), given the lack of available or reliable information on detailed future meteorological variables (Réveillet *et al*., 2018). The deep artificial neural network (ANN) approach is a promising new empirical approach to simulate SMB in the future (Bolibar *et al*., 2020). A neural network is a collection of interconnected simple processing elements called neurons. These processing elements are assigned coefficients or weights, which constitute the neural-network structure. Each weight is generated by the training process for the ANN (Agatonovic-Kustrin and Beresford, 2000).

Recently, Bolibar *et al*. (2022) analysed the sensitivity of glacier SMB to future climate change using a deep ANN. They write that, unlike linear statistical and temperature-index models, their deep-learning approach captures nonlinear responses of glacier SMB to high deviations in air temperature and solid precipitation, improving the representation of extreme SMBs. Bolibar *et al*. (2022) argue that temperature-index models, widely used to simulate the large-scale evolution of glaciers, provide only linear relationships between positive degree-days (PDDs), solid precipitation and SMB can be suitable for steep mountain glaciers, but may be less suitable for some scenarios and flatter glaciers and ice caps due to linear sensitivities in such mass balance models. Here, we performed numerical experiments with a classic and simple temperature-index model and the results demonstrated nonlinear responses of glacier SMB to temperature and precipitation changes, In this paper we perform numerical experiments with a classic and simple temperature-index model. Our unique purpose is to demonstrate that temperature-index models are able to capture nonlinear responses of glacier mass balance (MB) to high deviations in air temperature and solid precipitation.

**Data**

For our numerical experiments, we selected two very different glaciers in the French Alps. The first, the Argentière Glacier, is located in the Mont-Blanc range (45°55' N, 6°57'E). Its surface area was approximately 10.9 km² in 2018. The glacier extends from an altitude of approximately 3 400 m a.s.l. at the upper bergschrund down to 1 600 m a.s.l. at the snout. It faces north-west, except for a large part of the accumulation area (south-west facing tributaries). The second, the Sarennes Glacier, is a small south-facing glacier (0.51 km²) with a limited altitude range between 2 820 m and 3 160 m (mean values over

the period used for the present study), located in the Grande Rousses range (45°07'N; 6°07'E). The field SMB observations of the Argentière and Sarennes glaciers come from the French glacier monitoring program GLACIOCLIM (Les GLACIers, un Observatoire du CLIMat; https://glacioclim.osug.fr/). Annual SMBs were monitored in the ablation area of the Argentière Glacier between 1975 and 1993, using 20 to 30 ablation stakes. Since 1993, systematic winter and summer mass-balance measurements (May and September respectively) have been carried out over the entire surface of the glacier. Approximately 40 sites were selected at various elevations representative of the whole surface. Moreover, geodetic mass balances have been calculated using Digital Elevation Models on the basis of an old map from 1905 and photogrammetric measurements carried out in 1949, 1980, 1993, 1998, 2003, 2008 and 2019 (Vincent *et al.*, 2009). Since 1949, systematic winter and summer mass-balance measurements have been carried out on the Sarennes glacier, from which annual balances are calculated (Thibert *et al.*, 2013).

We used the atmospheric temperature and precipitation data from the SAFRAN (Système d'Analyse Fournissant des Renseignements Adaptés à la Nivologie, Analysis system for the provision of information for snow research) reanalysis process that are available from 1958 to date (Durand *et al.*, 2009; Verfaillie *et al.*, 2018). SAFRAN disaggregates large-scale meteorological analyses and observations in the French Alps. The analyses provide hourly meteorological data as a function of seven slope exposures (N, S, E, W, SE, SW and flat) and altitude (at 300 m intervals up to 3 600 m a.s.l), and that differ for each mountain range (e.g. Mont Blanc, Vanoise and Grandes Rousses ranges).

**Method**

We ran numerical experiments with a classic simple temperature-index model (Hock, 1999; Reveillet *et al.*, 2017) and using SAFRAN reanalysis data (Durand *et al.*, 2009; Verfaillie *et al.*, 2018). These numerical experiments were run on the two very different French glaciers, Argentière and Sarennes, observed over several decades (Thibert *et al.*, 2013; Vincent *et al.*, 2009). The SMB model was run for each day using the equation:

$SMB = DDF_{snow/ice} \cdot T + k \cdot P,$

Where:
-    T is the difference between the mean daily air temperature and the melting point,
-    $DDF_{snow/ice}$ is the degree-day factor for snow and ice and DDF=0 if T<0°C,
-    P is the precipitation (m w.e.),
-    *k* is a ratio between snow accumulation and precipitation and *k*=0 if T>0°C.
The degree-day factors for snow and ice were 0.0035 and 0.0055 m w.e. $K^{-1}d^{-1}$ for the Argentière glacier
(Reveillet *et al.*, 2017) and 0.0041 and 0.0068 m w.e. $K^{-1}d^{-1}$ for the Sarennes glacier (Thibert *et al.*,
2013). The point-mass balances were calculated for each elevation, for the Argentière and Sarennes
glaciers. In addition, we calculated the glacier-wide mass balance of the Argentière glacier using the
point-mass balances for the elevation range and the geodetic mass balances (Vincent *et al.*, 2009).
Parameter *k* depends on the site elevation in accounting for the precipitation gradient and is determined
from the winter-balance measurements and precipitation data.
Other enhanced temperature-index models including potential direct solar radiation could be used for
our study, but here the purpose is to show that responses in SMB are not linear to temperature or
precipitation changes even using a simple degree-day model.

**Results**
The reconstruction of the glacier-wide MBs of these glaciers from our simple temperature-index model
shows good agreement with data (Fig. 1). Using these reconstructed MBs, we calculated the SMB
sensitivities to temperature and winter precipitation at 2 750 metres and 3 100 metres on the Argentière
and Sarennes glaciers respectively (Fig. 2). These altitudes were selected because they correspond to
the approximate center of the glaciers. For each day of each series, we calculated an annual SMB
anomaly by adding a temperature anomaly or a precipitation anomaly. The anomaly was generated as a
shift (increment/decrement) of the mean of the distribution of the original data in temperatures and
winter balances. The distribution around the means was unchanged (same year-to-year variability as
found in the original data).
We report the results in Figure 2 to mirror Figure 3 of Bolibar *et al*. (2022) and make the comparison
easier. We also ran these numerical experiments at different altitudes and over the entire glacier surface
of the Argentière glacier (Fig. 3).
From our experiments, we found first that the response of SMB to temperature, using a temperature-
index model, is not linear~~, contrary to the conclusions of Bolibar *et al*. (2022) relative to temperature~~
~~index models~~. As expected, the sensitivity of annual SMB (i.e. the slope of the green curves in the graphs
of Figure 2) increases with the PDD anomaly. To explain the physical processes involved in
nonlinearity, we again used our PDD model, but using synthetic data for atmospheric temperature
changes over a year (Fig. 4a). The reference scenario (unforced temperature and winter-balance
reference conditions) of synthetic data is typical for a location in the upper ablation area of an Alpine
glacier (cumulative PDD of 800 degree.days from early May to early October; 1 700 mm of winter
balance). We use increments of $\pm 1K$ (-5K; +5K) to analyse the response of SMB. PDD factors for snow
and ice come from Thibert *et al.* (2013). As shown in Figure 4, the nonlinearity with respect to
temperature forcing (the spread between SMB plots in Fig.4c) comes from (i) the lengthening of the
ablation season (Fig.4a) and (ii) the earlier disappearance of the winter snow cover which increases the
ablation rate due to the change in the degree-day factor from snow to ice (Fig. 4b).
Concerning the winter balance, runs of our PDD model on synthetic data under different conditions of
winter balance (Fig. 5) used a reference scenario of 1 700 mm of winter balance changed by increments
of $\pm 300$ mm in precipitation. We found a nonlinear response of SMBs to winter precipitation with our
PDD model ~~and this is also inconsistent with the conclusions of Bolibar *et al*. (2022) relative to the~~
~~sensitivity of temperature index models~~. For instance, with winter accumulation decreased by -
1500 mm, ice ablation starts very early (by the end of May) and the annual MB is -5.55 m w.e. a$^{-1}$ in
October. With winter accumulation increased by +1500 mm, ice ablation starts in mid-September and
the annual MB is -0.21 m w.e. a$^{-1}$ in October. This asymmetry clearly shows that the response to winter
accumulation is not linear. Results show that the increase in sensitivity can be physically explained by
the earlier disappearance of the winter snow cover. The earlier and abrupt increase in the ablation rate
under lower conditions of winter balance (Fig.5a) results in nonlinearity attested by the spread between
SMB plots in Figure 5b. ~~Surprisingly, w~~We detect sensitivity to winter accumulation, contrary to the

Bolibar *et al.* (2022) findings using their ANN (Fig. 2 and 3). Indeed, MB sensitivity increases with low winter-accumulation anomalies using our model, but decreases in the deep-learning model of Bolibar *et al.* (2022). Our results are consistent with direct in-situ observations (Six and Vincent, 2014) and also consistent with the results reported by Reveillet *et al.* (2018) from observations and energy-balance modelling. The opposite results obtained from the deep-learning model ~~are paradoxical and~~ may be due to an issue in the calibration of the model.

Summing up, the ability of PDD models to provide nonlinear sensitivity to air temperature and solid precipitation is due to the different ablation rates and the associated change in the degree-day factor that can be involved depending on snow or ice conditions at the glacier surface. An additional nonlinearity to temperature forcing is caused by changes in the ablation duration.

Another question arises in the Discussion section of Bolibar *et al.* (2022), concerning the comparison between their results and those from other studies. ~~The authors claim that~~According to this paper, all glacier models in the Glacier Model Intercomparison Project (GlacierMIP) (Hock *et al*., 2019) rely on SMB models with linear relationships between PDDs, melt and precipitation. The authors argue that these PDD models present behaviour very similar to the linear-build statistical LASSO model. ~~This is erroneous given that,~~ However most of the temperature-index models used in GlacierMIP include two degree-day factors. Consequently, they cannot provide a linear response to climate forcing as shown above. In the Bolibar *et al.* (2022) paper, the MB anomalies in response to climate forcing were obtained using a linear LASSO SMB model, which is based on a regularized multi-linear regression, although~~.~~ ~~The choice of the LASSO model is even more surprising given that~~ the authors also used the GloGEMflow model in their paper (see their Discussion section), which is a temperature-index model widely used for glacier projections (Zekollari *et al*. 2019).

**Conclusions**

From ~~our~~ numerical experiments with a classic and simple temperature-index model, ~~we found nonlinear responses of glacier SMB to temperature and precipitation changes. These~~our results ~~question those of Bolibar *et al.* (2022), who argue that temperature index models provide only linear relationships between positive degree days (PDDs), solid precipitation and SMB.~~ highlight that temperature-index

models are able to capture nonlinear responses of glacier mass balance (MB) to high deviations in air temperature and solid precipitation, unlike Bolibar *et al.* (2022) study.

~~We tried to understand the cause of this discrepancy.~~ Bolibar *et al*. (2022) compared the response of SMB to climate forcing (air temperature, winter and summer snow falls) using a deep-learning approach and a LASSO model. From this comparison, they conclude that deep learning provides a nonlinear response, contrary to the LASSO model. The conclusions of Bolibar *et al*. (2022) may be due to the use of a linear LASSO SMB model instead of a temperature-index model. We would suggest testing the capability of an ANN to capture nonlinearity by comparing its results with that of the GloGEM Positive Degree-Day (PDD) model that they used in their paper.

Regarding specifically SMB changes due to solid precipitations, the deep-learning model used by Bolibar *et al*. (2022) foresees decreasing sensitivity under low winter-accumulation conditions. We point out that this result directly contradicts PDD model outcomes. We explain in physical terms why a PDD model projects higher sensitivity to low winter accumulation, but do not yet understand why the approach of Bolibar *et al*. (2022) does not.

Given that detailed meteorological variables are highly unpredictable in the future, most glacier-mass projections in response to climate change in large-scale studies spanning the $21^{st}$ century are still today based on temperature-index models with simple temperature and precipitation variables. It follows that the questions raised here relative to the nonlinear responses of surface SMB to meteorological variables are crucial.

**Data availability**

This commentary does not include original data. All data referred to in the text have been published elsewhere. Field data are accessible through the project website at https://glacioclim.osug.fr.

Results from the PDD simulations on synthetic data are accessible from the open data repository: 10.5281/zenodo.7603415.

**Author contributions**

ET and CV ran the numerical modelling calculations and produced the analysis. CV supervised the study
and wrote the paper. Both authors contributed to discussion of the results.

**Competing interests**
The authors declare that they have no conflicts of interest.

**Acknowledgements**
This study was funded by *Observatoire des Sciences de l'Univers de Grenoble* (OSUG) and *Institut des*
*Sciences de l'Univers* (INSU-CNRS) in the framework of the French GLACIOCLIM (*Les GLACIers,*
*un Observatoire du CLIMat*) program. We thank all those who conducted the field measurements. We
are grateful to Cary Bartsch for reviewing the English.

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

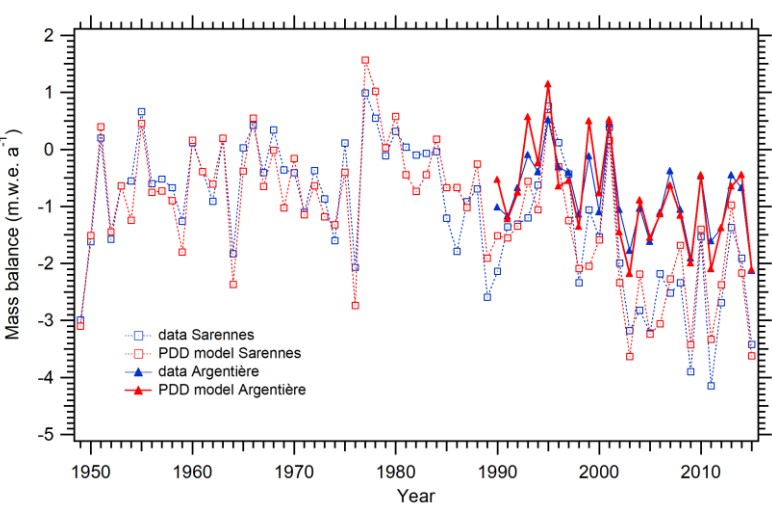


Figure 1. Glacier-wide mass balance of the Argentière glacier (1990-2015) and the Sarennes glacier

(1949-2015). Observations and simulations from the simple degree-day model used in our experiments.




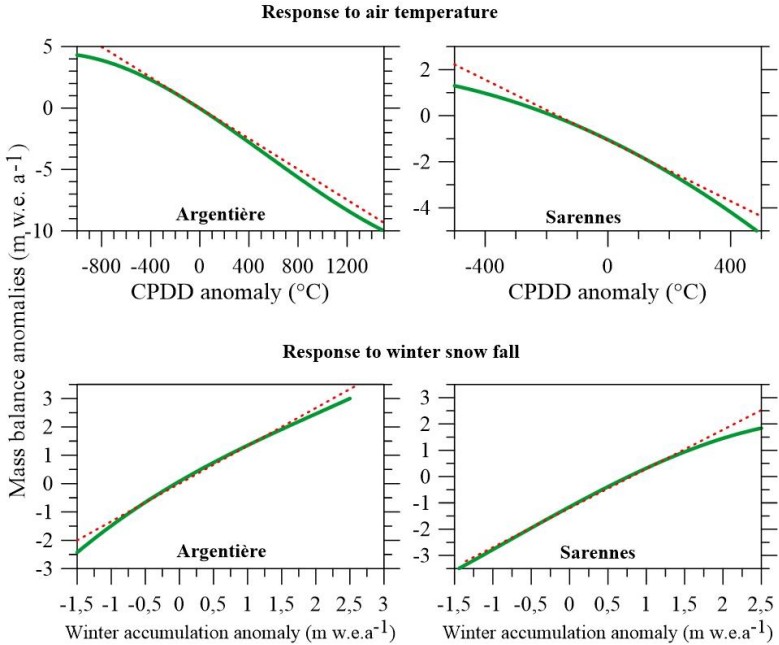


Figure 2. Response of mass balance to climate forcing using a temperature-index model (green line) at
2 750 m and 3 100 m on the Argentiere (left panel) and Sarennes (right panel) glaciers, respectively.
The red dashed lines are the best linear fit. Note that in such graphs, the sensitivity of the mass blance
to temperature and winter accumulation changes is the slope of the curves.

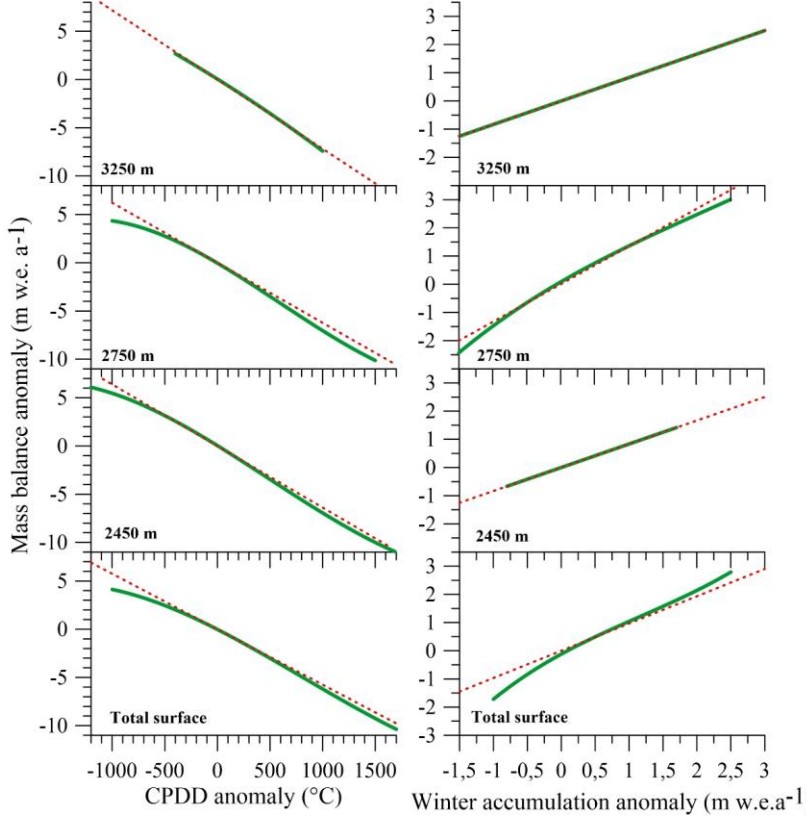



Figure 3. Response of annual mass balance to air temperature (left panel) and to winter accumulation
(right panel) using a temperature-index model (green line) on the Argentiere glacier. The red dashed
lines are the best fit forced through the origin.




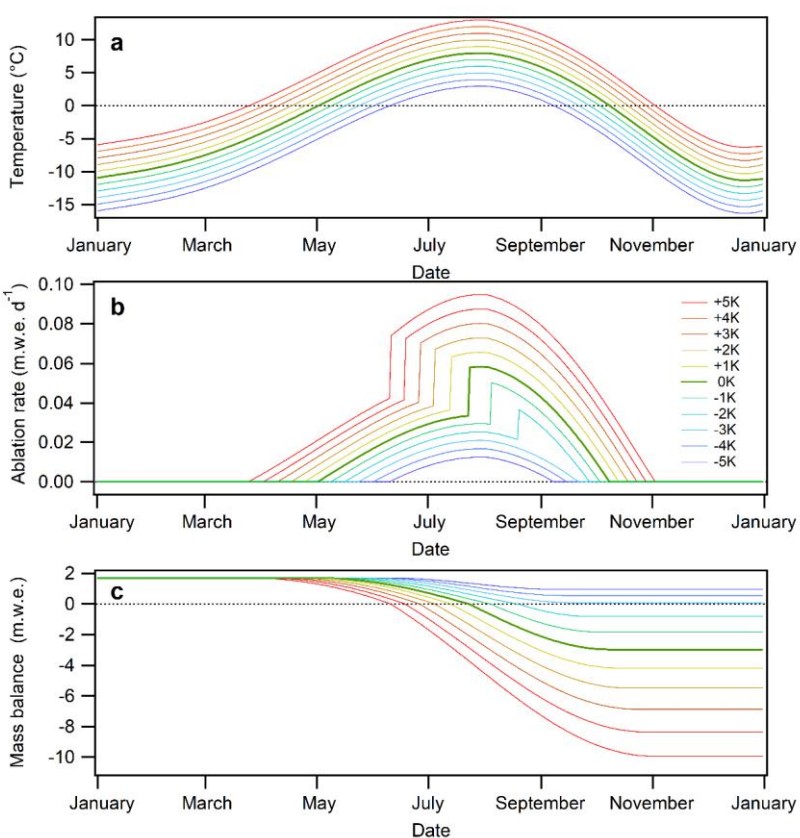


Figure 4. Positive degree-day model running on synthetic data (response to air temperature). Evolution

of air temperatures (a), ablation rates (b) and mass balance (c) over the year, according to different

temperature scenarios, calculated at 2 800 m. Note the jump in ablation rates when ablation shifts from

snow to ice. This occurs earlier with temperature forcing. Note also the lengthening of the ablation

season with rise in temperature.




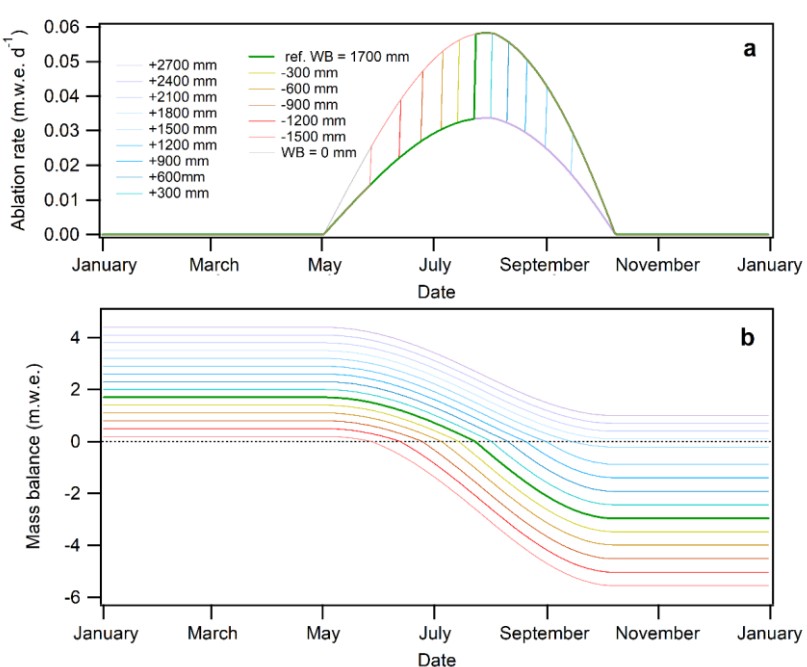


Figure 5. Positive degree-day model running on synthetic data (response to winter balance). Change in
ablation rates (a) and mass balance (b) over the year, according to different winter-balance scenarios
calculated at 2 800 m. Note the jump in ablation rates when ablation shifts from snow to ice. This occurs
earlier under lower winter-balance conditions. Note that the duration of the ablation season is unchanged
under variable winter-balance conditions.


