# Peer review of "Brief communication"

_The Cryosphere, 2022_

## Referee Comment (RC1)

**Review of "*Brief communication: Nonlinear sensitivity of glacier-mass balance attested by temperature-index models*" by *Vincent and Thibert**

The Cryosphere

**1 General comments**

Vincent and Thibert present a brief communication with an experiment based on two French alpine glaciers, which shows the response of point glacier mass balance at different glacier altitudes and glacier-wide mass balance to temperature and precipitation changes. This study is designed as a reply to the Nature Communications paper "Nonlinear sensitivity of glacier mass balance to future climate change unveiled by deep learning" by Bolibar et al. (2022)[1]. They claim that Bolibar et al. (2022) suggest that temperature-index models cannot capture nonlinear responses with respect to temperature and precipitation changes, and they aim at demonstrating the opposite.

This study serves as an extension to Bolibar et al. (2022), performing some additional analysis with temperature-index models that were not covered in that study. In that aspect, it serves to shed some additional light into the topic of glacier mass balance response to climatic changes. However, the scope of the study is very limited, and one is left feeling that only a few elements are analysed, often via cherry picking. My main concern regarding the study are the methods and the absence of objectivity in some of their claims. There is a lack of consistency in the way the information is presented and with which the different analyses are carried out. I will cover more in detail each one of these aspects in the following subsections of the global comments.

**1.1 GC1: Cherry picking of sentences out of context**

The first concern regarding this paper is the deliberate attempt to cherry pick sentences out of context in order to drive a point home. The most notorious of these is the widely repeated one in this study of "temperature-index models can only provide a linear relationship between positive degree-days (PDDs), solid precipitation and mass balance (MB)". While it is true that such a sentence is written in the article, many nuances are added around it. Bolibar et al. (2022) mention twice (pages 5 and 8), that the linear response to temperature is related to each individual degree-day factor (DDF), and that a temperature-index model with two DDFs (like the one in this study) virtually acts as a piece-wise function, able to partially account for some of the nonlinearities.

The authors seem very fixated with that idea, and seem to neglect this information altogether, showing a lack of objectivity. In that sense, this study serves to corroborate this hypothesis presented in Bolibar et al. (2022). Fig. 4c and Fig. 5 clearly display the piecewise behaviour of a temperature-index model with two DDFs.

In that sense, I believe it is important to nuance the message presented in this article, acknowledging the fact that this was already mentioned in Bolibar et al. (2022). This study is presented as an opposition to the message of Bolibar et al. (2022), whereas in fact it is building on top of it and corroborating a message evocated in that study. I would ask the authors to update all references to this sentence and to incorporate the elements described in this section into their study.

**1.2  GC2: Model calibration and validation**

Perhaps the most striking aspect of the methods is the lack of details regarding model calibration and validation. The authors present an equation used to model the mass balance, but they give no clues on how the two free parameters of the model (i.e. the DDFs of snow and ice) were obtained. The values are presented in the study, but one cannot know if these come from literature values or if these were calibrated somehow.

Seeing the model fit from Fig. 1, I am inclined to believe that these two parameters were manually calibrated, but it is unclear how that was performed.

- Is the model calibrated in an out-of-sample manner? Has the dataset been divided into a calibration/validation one and a test one? The model performance cannot be evaluated with the same data used for parameter calibration, otherwise one is ovefitting the model and reporting wrong metrics[2]. Please clearly explain how the model parameters have been calibrated, and if these have not been calibrated in an out-of-sample manner.

- What is the actual out-of-sample performance of the model for these two glaciers? Please report standard metrics (e.g. RMSE, bias and $r^2$)

- Have you taken into account ice dynamics in this model? How do you account for glacier geometry changes? How is the topographical feedback taken into account? Please specify.

- In Fig.1: why are there only MB simulations from 1990 onwards?

**1.3  GC3: Interpretation of the glacier mass balance nonlinearities**

An important aspect regarding this study is the interpretation of the nonlinear response of glacier mass balance to different climatic drivers (air temperature and winter snowfall in this case). The authors correctly point out that the reason behind the nonlinearities captured by their temperature-index model are the changes in duration of the accumulation and ablation season, which impact the snow/ice coverage ratio. While this is indeed one of the multiple nonlinear effects present in the response of glacier MB to climatic drivers, it is not the only one. The global picture is much more complex than that, with a complex combination of multiple feedbacks. These nonlinear effects are linked to the non stationarity of model parameters (i.e. DDFs for snow, firn and ice) in both the temporal and spatial dimensions[3]. These can vary in

magnitude, and depending on the topographical and climatic setup of each glacier, one might be more important than the other. From our current understanding of these processes, these are the main ones:

1. **The influence of variations of the surface energy budget components under climate change**: This was the main topic of discussion and the most important result in Bolibar et al. (2022). Since the role of shortwave radiation in the energy budget in the past (i.e. the calibration period) is higher than in the future under climate change, its importance is bound to decrease in the future[4]. This results in a REDUCED sensitivity of DDFs (particularly of ice, due to its lower albedo) to future warming. This corroborates many studies in the literature that also encountered an overestimation of DDFs sensitivity to future warming[3,5,6]. For the whole region of the French Alps, Bolibar et al. (2022) found that this was the main nonlinear effect, driving differences in projected mass balance changes. Nonetheless, Bolibar et al. (2022) found that this was true only for glaciers with long response times or flat glaciers, due to the reduced effect of topographical adjustment. This nonlinearity affects parameters in the TEMPORAL dimension, resulting in a decrease in sensitivity over time, as air temperature rises.

2. **The influence of different surface types and therefore different DDFs in the temperature-index model**: The use of multiple DDFs for snow, firn and ice results in a nonlinearity in the SPATIAL domain. This nonlinear response will be affected by the spatial distribution of snow, firn and ice over the glacier. This spatial distribution will indeed also change through time, which will determine the switch between DDFs in the ablation season. Nonetheless, it is highly tied to glacier hypsometry. As reported in this study, in a warming climate, this lengthening of the ablation season exposes more ice surface linked to higher DDFs and therefore INCREASES the sensitivity.

3. **Surface albedo**: Changes in surface albedo through time also introduce a nonlinear response to warming in the TEMPORAL domain. These are also linked to 1, but they produce an opposite effect. Generally, in a warmer climate, surfaces tend to darken, thus further INCREASING the sensitivity of DDFs[7]. As mentioned, this process works in opposition to 1, so depending on the different topo-climatic setups, one might become more important than the other one.

4. **Glacier hypsometry**: This one affects both the previous processes, and it serves to display how complex are the interactions between all these feedbacks. As explained in Bolibar et al. (2022), flatter glaciers or glaciers with a long response time will display less topographical adjustment, thus enduring more extreme air temperatures. This will result in more climatic extremes and therefore increased nonlinear effects due to the reduced influence of shortwave radiation. Therefore, flatter glaciers will tend to display REDUCED senstivities to warming, whereas steep glaciers will not see many differences.

   The results of this study help shed light on the above-metioned point 2, but one should not jump too quickly to conclusions just because a model does display nonlinearities. As I just tried to argue, these nonlinear responses are combined in complex manners, and they

are not straightforward to disentangle. While Bolibar et al. (2022) found that the above-mentioned point 1 seemed to be the most important nonlinear effect for the French Alps, this will most likely vary depending on the region and climate scenarios. More studies are needed to try to disentangle these nonlinear effects and to better understand their importance and effects for a wide range of topo-climatic setups.

In that sense, I believe it is important to mention and take into account this global picture in the conclusions of this study. Therefore, I think the results related to these nonlinear response should be presented as one of the multiple nonlinear responses of MB to climate change. Temperature-index models can indeed partially capture as a piece-wise function nonlinear effects linked to the spatial domain, but it remains unclear which is the most important nonlinear effect for multiple glaciological regions. Framing the results in this wider context will help place the scientific contributions of this study into the big picture.

**1.4 GC4: Summer snowfall anomalies and plotting of nonlinear response**

Bolibar et al. (2022) encountered that the strongest nonlinear response (from a statistical point of view) came from summer snowfall anomalies. The authors argued that it was the combination of both air temperature and precipitation during summer that determined wide changes in MB sensitivity. As explained above, Bolibar et al. (2022) argue that this is due to a reduced role of short-wave radiation in future climate scenarios, resulting in a reduced sensitivity of DDFs. Summer snowfall anomalies are tightly linked to summer air temperatures and also the ratio between snow and ice coverage on a glacier. These two are closely linked to processes mentioned in point 1 and 2 above, and were found to be the clearest drivers of nonlinearities. The statistical methods of Bolibar et al. (2022) served to shed some additional light on the subject, and open the door to exploring new ways to disentangle these processes. However, they did not allow a clear separation and understanding of how these processes operate.

Another important aspect in the comparison between the nonlinear sensitivities of Fig. 3 in Bolibar et al. (2022) with respect to Fig. 2 of this study, is the use of equivalent axis and ranges of values. Right now, both figures do not share the same range of values, and as it was displayed in Fig. 3 of Bolibar et al. (2022), there is a reduced range of values that will be encountered by French Alpine glaciers in future climate scenarios for different RCPs. This is particularly problematic for the case of winter snowfall anomalies. In Fig. 2 of this study, slight nonlinearities are displayed below -1.2 m.w.e. and above +1.7 m.w.e. These values are way beyond anything that will be seen in the 21st century for French alpine glaciers, as displayed in the vertical dashed lines in Fig. 3 of Bolibar et al. (2022). The most extreme values that French alpine glaciers will see until 2100 will range between -0.7 m.w.e. to +1.2 m.w.e. Anything beyond these limits makes no sense from a physical point of view for this sort of analyses, and will have no impact in projections for this century. At the very least, Vincent and Thibert should admit that nonlinearities of MB shown by Bolibar et al. (2022) for very extreme anomaly values out of the range of future likely encountered values, must simply not be taken into account in their analyses.

Additionally, one aspect that is not mentioned in this study is the fact that they are comparing the response of two glaciers with that of 660 glaciers. Bolibar et al. (2022) reported

a strong variability in terms of mass balance sensitivity response to climatic forcings along different types of glaciers. The very reduced sampling used by Vincent and Thibert shows just a partial picture of all glaciers in the region. This should be specifically mentioned when presenting the comparisons.

In order to better understand these effects and to better compare both methods, I believe it is necessary to add the response of summer snowfall anomalies to Fig. 2. This would allow a comparison with the most meaningful response of the methods of Bolibar et al. (2022). Moreover, the future ranges of extreme values encountered by these glaciers under future climate scenarios (e.g. using the ADAMONT[8] product which is compatible with the SAFRAN[9] product used in this study), should be added to Figs. 2 and 3. This would clearly indicate where the nonlinearities actually will come into play and where they will be just model extrapolations beyond physically plausible values. This would also show that the nonlinearities linked to winter snowfall anomalies illustrated in Fig. 3 of Bolibar et al.(2022) will never occur during the 21st century in the French Alps, as they are out of the range of the values simulated by climate models.

**1.5  GC5: Code and data availability**

Another aspect that makes it hard to understand the methods is the fact that the source code used for this study is not shared. Following the principles of open science from The Cryosphere journal, I would strongly encourage the authors to share their code and data in an open repository (e.g. GitHub). This would make the study reproducible, and it would make it easier for reviewers and readers to understand what has been done.

If the authors strongly oppose to this, I would still ask them to privately share their code for this review in order to correctly understand what has been done.

**2  Specific comments**

- **L1** The current title does not give much information on what the sensitivity is linked to. I believe a correct title should be something like "Nonlinear sensitivity of glacier mass balance to future climate change attested by temperature-index models".

- **L18** The aspects regarding GC3 should be added here in the abstract.

- **L33-35** This is one of the cherry picking instances mentioned in GC1. To be adjusted accordingly.

- **L53** How has the temperature been downscaled to be used in the temperature-index model? Two versions of SAFRAN exist: one divided by massifs and altitudinal bands, and another one in a grid. Which one of the two has been used?

- **L61** As per the comments on GC2: how have been these two DDFs been obtained?

- **L79** This is one of the cherry picking instances mentioned in GC1. To be adjusted accordingly.

- **L97-98** This sentence is lacking solid arguments to back it. Could you please elaborate? Which parts of the model calibration might have issues? All the models in Bolibar et al. (2022) were cross-validated, ensuring a correct out-of-sample validation and a good generalization outside the seen dataset. This study so far does not provide any information regarding parameter calibration. In order to correctly compare both models and draw conclusions, a good understanding of both model calibration strategies is necessary.

- **L99-101** This is indeed true, and has already been reported in other studies. However, as argued in Bolibar et al. (2022) and as I explained in GC3, this is only part of the picture. This should be adjusted to mention that this is one of the multiple nonlinear processes in glacier mass balance sensitivity to climatic forcing, and that this process is a different one that the one reported in Bolibar et al. (2022).

- **L104-106** This is again a case of cherry picking. Bolibar et al. (2022) never claimed that ALL models in GlacierMIP 2[10] (not GlacierMIP 1, Hock et al. (2019), as stated by the authors) have linear relationships to PDDs and precipitation. To begin with, some of them use SEB. Bolibar et al.(2022) make a point that temperature-index models with a single DDF clearly behave like the Lasso; and even temperature-index models with 2 DDFs, can only partially account for the nonlinearities (and cannot capture the ones they show in their study). This is further corroborated by the comparisons made in that study between the Lasso MB model and the temperature-index MB model from GloGEM in the Supplementary material of Bolibar et al. (2022). To be modified accordingly.

- **L107-108** This is not accurate. The Open Global Glacier Model (OGGM), which was used in the paper[1] as an example of this behaviour, also has a single DDF.

  In order to avoid further cherry picking, I would ask the authors to be precise about their claims. Out of the of the 11 models in Marzeion et al. (2020)[10], 7 are using temperature-index models (2 of them with a single DDF), 1 is using a simple parametrizations relating MB to air temperature, 1 is using a mass balance gradient based on temperature indices, and 2 are using surface energy balance models. This means that at least 3 (potentially 4 if we take into account Kraaijenbrink et al. (2017)) models have direct simple linear relationships between PDDs and MB. The other 4 have 2 DDFs, which can partially account for nonlinearities (but not the ones from the above-mentioned point 1 in the temporal dimension).

- **L117-119** This is one of the cherry picking instances mentioned in GC1. To be adjusted accordingly.

- **L124-126** This was already done in that study. The results were shown in the Supplementary material. The exact same plots were not produced due to the difficulty of implementing that scheme on GloGEM. But the evolution of the MB for future scenarios was compared, yielding very similar results and responses to those of the LASSO. Therefore, the comparison between the LASSO and the TI model from GloGEM in terms of projected cumulative MB is not unfounded. Vincent and Thibert must point this aspect in an objective manner.

- **L129-131** As previously discussed in GC3, this is because the TI model used in this study does not account for DDF evolution over time. To be mentioned here in order to clarify the bigger picture.

**References**

1. Bolibar, J., Rabatel, A., Gouttevin, I., Zekollari, H. & Galiez, C. Nonlinear sensitivity of glacier mass balance to future climate change unveiled by deep learning. en. *Nature Communications* **13,** 409. ISSN: 2041-1723. https://www.nature.com/articles/s41467-022-28033-0 (2022) (Dec. 2022).

2. Hastie, T., Tibshirani, R. & Friedman, J. *The Elements of Statistical Learning* ISBN: 978-0-387-84857-0 978-0-387-84858-7. http://link.springer.com/10.1007/978-0-387-84858-7 (2019) (Springer New York, New York, NY, 2009).

3. Ismail, M. F., Bogacki, W., Disse, M., Schäfer, M. & Kirschbauer, L. Estimating degree-day factors based on energy flux components. *The Cryosphere Discussions* **2022,** 1–40. https://tc.copernicus.org/preprints/tc-2022-64/ (2022).

4. Huss, M., Funk, M. & Ohmura, A. Strong Alpine glacier melt in the 1940s due to enhanced solar radiation. en. *Geophysical Research Letters* **36,** L23501. ISSN: 0094-8276. http://doi.wiley.com/10.1029/2009GL040789 (2021) (Dec. 2009).

5. Braithwaite, R. J. Positive degree-day factors for ablation on the Greenland ice sheet studied by energy-balance modelling. en. *Journal of Glaciology* **41,** 153–160. ISSN: 0022-1430, 1727-5652. https://www.cambridge.org/core/product/identifier/S0022143000017846/type/journal_article (2021) (1995).

6. Matthews, T., Perry, L., Guy, H., Wilby, R. & Edwards, T. *Modelling the impact of atmospheric heat accumulation on glacier mass balance* preprint (In Review, Oct. 2022). https://www.researchsquare.com/article/rs-2166876/v1 (2022).

7. Naegeli, K. & Huss, M. Sensitivity of mountain glacier mass balance to changes in bare-ice albedo. en. *Annals of Glaciology* **58,** 119–129. ISSN: 0260-3055, 1727-5644. https://www.cambridge.org/core/product/identifier/S0260305517000258/type/journal_article (2021) (July 2017).

8. Verfaillie, D., Déqué, M., Morin, S. & Lafaysse, M. The method ADAMONT v1.0 for statistical adjustment of climate projections applicable to energy balance land surface models. en. *Geoscientific Model Development* **10,** 4257–4283. ISSN: 1991-9603. https://www.geosci-model-dev.net/10/4257/2017/ (2018) (Nov. 2017).

9. Durand, Y. *et al.* Reanalysis of 44 Yr of Climate in the French Alps (1958–2002): Methodology, Model Validation, Climatology, and Trends for Air Temperature and Precipitation. en. *Journal of Applied Meteorology and Climatology* **48,** 429–449. ISSN: 1558-8424, 1558-8432. http://journals.ametsoc.org/doi/abs/10.1175/2008JAMC1808.1 (2019) (Mar. 2009).

10. Marzeion, B. *et al.* Partitioning the Uncertainty of Ensemble Projections of Global Glacier Mass Change. en. *Earth's Future.* ISSN: 2328-4277, 2328-4277. `https://onlinelibrary.wiley.com/doi/abs/10.1029/2019EF001470` (2020) (Apr. 2020).

---

## Author Comment (AC1)

**Reply to Reviewers :**

*We have complied with all the comments made by Reviewers (see the point-by-point response). The replies are in italics.*

*The manuscript has been revised by a professional native speaker (not this Reply).*

**Reply to Reviewer 1 (J. Bolibar):**

*First, we would like to remind kindly the obligations for a referee as mentioned in the following paragraphs of policies of The Cryosphere journal:*

*« 4. A referee should be sensitive even to the appearance of a conflict of interest when the manuscript under review is closely related to the referee's work in progress or published. If in doubt, the referee should return the manuscript promptly without review, advising the editor of the conflict of interest or bias.*
*5.       A referee should not evaluate a manuscript authored or co-authored by a person with whom the referee has a personal or professional connection if the relationship has the potential to bias judgement of the manuscript. »*

*One can conclude that there is here a clear conflict of interest.*

*However, we have complied with all the comments made by Reviewer 1, below.*

 **General comments** :

Vincent and Thibert present a brief communication with an experiment based on two French alpine glaciers, which shows the response of point glacier mass balance at different glacier altitudes and glacier-wide mass balance to temperature and precipitation changes. This study is designed as a reply to the Nature Communications paper "Nonlinear sensitivity of glacier mass balance to future climate change unveiled by deep learning" by Bolibar et al. (2022). They claim that Bolibar et al. (2022) suggest that temperature-index models cannot capture nonlinear responses with respect to temperature and precipitation changes, and they aim at demonstrating the opposite.
This study serves as an extension to Bolibar et al. (2022), performing some additional analysis with temperature-index models that were not covered in that study. In that aspect, it serves to shed some additional light into the topic of glacier mass balance response to climatic changes. However, the scope of the study is very limited, and one is left feeling that only a few elements are analysed, often via cherry picking. My main concern regarding the study are the methods and the absence of objectivity in some of their claims. There is a lack of consistency in the way the information is presented and with which the different analyses are carried out. I will cover more in detail each one of these aspects in the following subsections of the global comments.

**1.1 General comment 1: Cherry picking of sentences out of context**

The first concern regarding this paper is the deliberate attempt to cherry pick sentences out of context in order to drive a point home. The most notorious of these is the widely repeated one in this study of "temperature-index models can only provide a linear relationship between positive degree-days (PDDs), solid precipitation and mass balance (MB)". While it is true that such a sentence is written in the article, many nuances are added around it. Bolibar et al. (2022) mention twice (pages 5 and 8), that the linear response to temperature is related to each individual degree-day factor (DDF), and that a temperature-index model with two DDFs (like the one in this study) virtually acts as a piece-wise function, able to partially account for some of the nonlinearities.  The authors seem very fixated with that idea, and seem to neglect this information altogether, showing a lack of objectivity. In that sense, this study serves to corroborate this hypothesis presented in Bolibar et al. (2022). Fig. 4c and Fig. 5 clearly display the piecewise behaviour of a temperature-index model with two DDFs. In that sense, I

believe it is important to nuance the message presented in this article, acknowledging the fact that this was already mentioned in Bolibar et al. (2022). This study is presented as an opposition to the message of Bolibar et al. (2022), whereas in fact it is building on top of it and corroborating a message evocated in that study. I would ask the authors to update all references to this sentence and to incorporate the elements described in this section into their study.

*First, we recognize the novelty of the approach of the authors. We are convinced that the deep artificial neural network (ANN) approach is a promising new empirical approach to simulate surface MB in the future, as it has been done in Bolibar et al. (2020) and in Bolibar et al. (2022).*
*Our unique purpose is to show that temperature-index models are able to capture nonlinear responses of glacier mass balance (MB) to high deviations in air temperature and solid precipitation.*
*Bolibar et al. (2022) argue that temperature-index models provide only linear relationships between positive degree-days (PDDs), solid precipitation and MB. Our statements do not result from"cherry picking" as shown below.*

*1)      First, the title of Bolibar et al (2022): « Non linear sensitivity of glacier mass balance to future climate change unveiled by deep learning » implies that previous approaches at hands (not only temperature index-based approaches) were linear in comparison to the new ANN approach.*

*2)      In the abstract of Bolibar et al. (2022), one can read : « Deep learning captures a nonlinear response of glaciers to air temperature and precipitation, improving the representation of extreme mass balance rates compared to linear statistical and temperature-index models. Our results confirm an over-sensitivity of temperature-index model, often used by large-sacle studies, to future warming. » . These lines correspond to 4 lines of Abstract (or 1/3 of the Abstract) and we do not think this falls under « cherry picking ».*

*3)      In the Introduction of Bolibar et al. (2022), on can read : « This type of model uses a calibrated linear relationship between positive degree-days (PDDs) and the melt of ice or snow. The main reason for their success comes from their suitability to large-scale studies with a low density of observations, in some cases displaying an even better performance than more complex models. However, both the climate and glacier systems are known to react non-linearly, even to pre-processed forcings like PDDs, implying that these models can only offer a linearized approximation of climate-glacier relationships. »*

*4)       In the results section of Bolibar et al. (2022), on can read : « In that study, a temperature-index model with a separate degree-day factor (DDF) for snow and ice is used, resulting in piecewise linear functions able to partially reproduce nonlinear MB dynamics ». However, they write in the following sentences: « Both the Lasso and the temperature-index MB model rely on linear relationships between PDDs, solid precipitation and MB. Therefore, their sensitivities to the projected 21st century increase in PDDs are linear. »*

*5)      In Discussion of Bolibar et al. (2022), the authors justify the analogy between Lasso model and Temperature index model. They write : « At this point, it is important to clarify the different ways of treating PDDs in the Lasso and the temperature-index MB models analysed in this study in order to justify analogies » and they conclude : « Nonetheless, since they are both linear, their calibrated parameters establishing the sensitivity of melt and glacier-wide MB to temperature variations remain constant over time. »*

*6)      When they analyse the glacier models in GlacierMIP, they bring some nuances:*
*« Despite the existence of a wide variety of different approaches to simulate glacier dynamics, all glacier models in GlacierMIP rely on MB models with linear relationships between PDDs and melt, and precipitation and accumulation. Some of these models use a single DDF, while others have separate DDFs for snow and ice, producing a piecewise function composed of two linear sub-functions that can partially account for nonlinear MB dynamics.", but*

*they finally conclude that « As we have previously shown, these models present a very similar behaviour to the linear statistical MB model from this study » (i.e Lasso model)*

*7) And in the last paragraph of Bolibar et al. (2022), the authors claim that « By unravelling nonlinear relationships between climate and glacier MB, we have demonstrated the limitations of linear statistical MB models to represent extreme MB rates in long-term projections. Our analyses suggest that these limitations can also be translated to temperature-index MB models, as they share linear relationships between PDDs and melt, as well as precipitation and accumulation »*

*Thus, the assertion of the linear behaviour of TI models is reitered 7 times. We do not believe that we attempted "to cherry pick sentences out of context"*

*Reviewer 3 of our study wrote the same thing : « The Abstract of Bolibar et al. (2022) states twice very clearly that temperature-index models have a linear sensitivity. And even the title ("nonlinear sensitivity ... unveiled ...") indirectly implies that previous approaches were linear in comparison to the new model. Even though the text indeed provides additional statements that actually better agree with the outcomes of this study, it is the Title and the Abstract that defines what readers take with them. »*

*However, again, our purpose is not to criticize the ANN approach used in Bolibar et al. (2022). Our purpose is to demonstrate–as far as this is not a novelty– the ability of temperature-index models to capture nonlinear responses of mass balance to temperature and precipitation changes. This would be a detail if the most glacier-mass projections in response to climate change in large-scale studies over the 21st century were not based on temperature-index models.*

**1.2 GC2: Model calibration and validation**

Perhaps the most striking aspect of the methods is the lack of details regarding model calibration and validation. The authors present an equation used to model the mass balance, but they give no clues on how the two free parameters of the model (i.e. the DDFs of snow and ice) were obtained. The values are presented in the study, but one cannot know if these come from literature values or if these were calibrated somehow. Seeing the model fit from Fig. 1, I am inclined to believe that these two parameters were manually calibrated, but it is unclear how that was performed. • Is the model calibrated in an out-of-sample manner? Has the dataset been divided into a calibration/validation one and a test one? The model performance cannot be evaluated with the same data used for parameter calibration, otherwise one is ovefitting the model and reporting wrong metrics . Please clearly explain how the model parameters have been calibrated, and if these have not been calibrated in an out-of-sample manner. • What is the actual out-of-sample performance of the model for these two glaciers? Please report standard metrics (e.g. RMSE, bias and r 2 ) • Have you taken into account ice dynamics in this model? How do you account for glacier geometry changes? How is the topographical feedback taken into account? Please specify. • In Fig.1: why are there only MB simulations from 1990 onwards?

*We agree that a little bit more can be said about the methods although these PDD models are widely used in the glaciological community. More details about mass balance, DDF calibration and DEM data are now given in the Data section. Point mass balances are calculated for each elevation, for Argentière and Sarennes glaciers. In addition, we calculated the glacier-wide mass balance of Argentière glacier using the point mass balances for elevation range and geodetic mass balances (Vincent et al., 2009).*
*The degree-day factors for snow and ice are 0.0035 and 0.0055 m w.e. $K^{-1}d^{-1}$ for Argentière glacier (Reveillet et al., 2017) and 0.0041 and 0.0068 m w.e. $K^{-1}d^{-1}$ for Sarennes glacier (Thibert et al., 2013). The calibration and validation of these factors have been done in (Reveillet et al., 2017) and (Thibert et al., 2013). It is not possible to provide more details in the present paper but further information can be found in these papers. Except if the degree-day factors for snow and ice are the same, the response*

*of temperature index model is not linear and different pairs of degree-day factors do not change the conclusions, as shown with the numerical experiments performed from synthetic data (Fig. 4 and 5).*
*In order to estimate the response of MB calculated from TI model, it is better to analyse the response of point mass balance, in order to get rid of dynamics feedbacks. We also ran numerical experiments over the entire glacier surface and found similar results (Fig. 3).*
*The MB simulations of Argentiere glacier (Fig. 1) start in 1990 because the observed winter, summer and annual mass balances are not available before 1990 over the whole surface of the glacier. The data performed before 1990 have been reconstructed from annual mass balance observations performed in the ablation area and a statistical model (Vincent et al., 2018). In addition, the data performed before 1990 do not correspond always to the end of ablation season. For the present study, we prefer to use observations which are not affected by potential large uncertainties.*

*Vincent, C., Soruco, A., Azam, M. F., Basantes-Serrano, R., Jackson, M., Kjøllmoen, B., et al. (2018). A nonlinear statistical model for extracting a climatic signal from glacier mass balance measurements. Journal of Geophysical Research: Earth Surface, 123. https://doi. org/10.1029/2018JF004702*

**1.3 **GC3: Interpretation of the glacier mass balance nonlinearities**

An important aspect regarding this study is the interpretation of the nonlinear response of glacier mass balance to different climatic drivers (air temperature and winter snowfall in this case). The authors correctly point out that the reason behind the nonlinearities captured by their temperature-index model are the changes in duration of the accumulation and ablation season, which impact the snow/ice coverage ratio. While this is indeed one of the multiple nonlinear effects present in the response of glacier MB to climatic drivers, it is not the only one. The global picture is much more complex than that, with a complex combination of multiple feedbacks. These nonlinear effects are linked to the non stationarity of model parameters (i.e. DDFs for snow, firn and ice) in both the temporal and spatial dimensions. These can vary in magnitude, and depending on the topographical and climatic setup of each glacier, one might be more important than the other. From our current understanding of these processes, these are the main ones.

1.The influence of variations of the surface energy budget components under climate change: This was the main topic of discussion and the most important result in Bolibar et al. (2022). Since the role of shortwave radiation in the energy budget in the past (i.e. the calibration period) is higher than in the future under climate change, its importance is bound to decrease in the future . This results in a REDUCED sensitivity of DDFs (particularly of ice, due to its lower albedo) to future warming. This corroborates many studies in the literature that also encountered an overestimation of DDFs sensitivity to future warming. For the whole region of the French Alps, Bolibar et al. (2022) found that this was the main nonlinear effect, driving differences in projected mass balance changes. Nonetheless, Bolibar et al. (2022) found that this was true only for glaciers with long response times or flat glaciers, due to the reduced effect of topographical adjustment. This nonlinearity affects parameters in the TEMPORAL dimension, resulting in a decrease in sensitivity over time, as air temperature rises.

2. The influence of different surface types and therefore different DDFs in the temperature-index model: The use of multiple DDFs for snow, firn and ice results in a nonlinearity in the SPATIAL domain. This nonlinear response will be affected by the spatial distribution of snow, firn and ice over the glacier. This spatial distribution will indeed also change through time, which will determine the switch between DDFs in the ablation season. Nonetheless, it is highly tied to glacier hypsometry. As reported in this study, in a warming climate, this lengthening of the ablation season exposes more ice surface linked to higher DDFs and therefore INCREASES the sensitivity.

3. Surface albedo: Changes in surface albedo through time also introduce a nonlinear response to warming in the TEMPORAL domain. These are also linked to 1, but they produce an opposite effect. Generally, in a warmer climate, surfaces tend to darken, thus further INCREASING the sensitivity of DDFs7 . As mentioned, this process works in opposition to 1, so depending on the different topo-climatic setups, one might become more important than the other one.

4. Glacier hypsometry: This one affects both the previous processes, and it serves to display how complex are the interactions between all these feedbacks. As explained in Bolibar et al. (2022), flatter glaciers or glaciers with a long response time will display less topographical adjustment, thus enduring more extreme air temperatures. This will result in more climatic extremes and therefore increased nonlinear effects due to the reduced influence of shortwave radiation. Therefore, flatter glaciers will tend to display REDUCED senstivities to warming, whereas steep glaciers will not see many differences. The results of this study help shed light on the above-metioned point 2, but one should not jump too quickly to conclusions just because a model does display nonlinearities. As I just tried to argue, these nonlinear responses are combined in complex manners, and they 3 are not straightforward to disentangle. While Bolibar et al. (2022) found that the abovementioned point 1 seemed to be the most important nonlinear effect for the French Alps, this will most likely vary depending on the region and climate scenarios. More studies are needed to try to disentangle these nonlinear effects and to better understand their importance and effects for a wide range of topo-climatic setups. In that sense, I believe it is important to mention and take into account this global picture in the conclusions of this study. Therefore, I think the results related to these nonlinear response should be presented as one of the multiple nonlinear responses of MB to climate change. Temperature-index models can indeed partially capture as a piece-wise function nonlinear effects linked to the spatial domain, but it remains unclear which is the most important nonlinear effect for multiple glaciological regions. Framing the results in this wider context will help place the scientific contributions of this study into the big picture.

*Agree. We are convinced that there are many sources of nonlinear effects and this point has been widely discussed in the literature (e.g. Oerlemans and Klok, 2002; MacDougall and Flowers, 2011). However, it is beyond the scope of our study to discuss the numerous potential other causes of nonlinear response of mass balance to climate change. Our present study just aimed to prove that a simple temperature index-model is able to reproduce nonlinear responses of glacier mass balance to temperature and precipitation.*

*The limitations of the temperature index-models are obvious and widely discussed in previous studies (Huss et al., 2009; Gabbi et al., 2014; Réveillet et al., 2017). Among these limitations, one can note (i) the temporal variations of melt sensitivity to temperature and (ii) the fact that the physical link between temperature and melt is not direct...*

*Physical approaches which consider all energy exchanges between the glacier and the atmosphere and are able to represent snow melt spatial and temporal variability, such as those related to albedo variations that are hard to represent in Temperature index models. Such approaches offer higher transferability over time (e.g., MacDougall and Flowers, 2011) but require more accurate meteorological forcing (e.g., Gabbi et al., 2014).*

*Given the lack of available or reliable information on detailed future meteorological variables, most glacier-mass projections in response to climate change in large-scale studies over the 21$^{st}$ century have been based on temperature-index models (Huss and Hock, 2015; Fox-Kemper et al., 2021),*

*It is crucial to note the limitations of the temperature index models but it is also crucial to recognise that temperature-index models are able to capture nonlinear responses of mass balance to temperature and precipitation. This is the only topic of our study.*

*Gabbi, J. M. Carenzo, F. Pellicciotti, A. Bauder and M. Funk (2014) A comparison of empirical and physically based glacier surface melt models for long-term simulations of glacier response. J. Glaciol., 60, 1140–1154. doi:10.3189/2014JoG14J011*

*Huss, M., M. Funk and A Ohmura (2009), Strong Alpine glacier melt in the 1940s due to enhanced solar radiation. Geophys. Res. Lett., 36. doi:10.1029/2009GL040789*

*MacDougall, A. H., and G. E. Flowers (2011), Spatial and Temporal Transferability of a Distributed*

*Energy-Balance Glacier Melt Model, J. of Climate, 24(5), 1480–1498, doi :10.1175/2010JCLI3821.1.*

*Oerlemans, J., and E. J. Klok (2002), Energy Balance of a Glacier Surface: Analysis of Automatic Weather Station Data from the Morteratschgletscher, Switzerland. Arct. Antarct. Alp. Res., 34, 477. doi:10.2307/1552206*

*Réveillet, M., C. Vincent, D. Six and A. Rabatel, A (2017), Which empirical model is best suited to simulate glacier mass balances? J. Glaciol., 63(237), 39-54. doi:10.1017/jog.2016.110*

**1.4 GC4: Summer snowfall anomalies and plotting of nonlinear response**

Bolibar et al. (2022) encountered that the strongest nonlinear response (from a statistical point of view) came from summer snowfall anomalies. The authors argued that it was the combination of both air temperature and precipitation during summer that determined wide changes in MB sensitivity. As explained above, Bolibar et al. (2022) argue that this is due to a reduced role of short-wave radiation in future climate scenarios, resulting in a reduced sensitivity of DDFs. Summer snowfall anomalies are tightly linked to summer air temperatures and also the ratio between snow and ice coverage on a glacier. These two are closely linked to processes mentioned in point 1 and 2 above, and were found to be the clearest drivers of nonlinearities. The statistical methods of Bolibar et al. (2022) served to shed some additional light on the subject, and open the door to exploring new ways to disentangle these processes. However, they did not allow a clear separation and understanding of how these processes operate.

Another important aspect in the comparison between the nonlinear sensitivities of Fig. 3 in Bolibar et al. (2022) with respect to Fig. 2 of this study, is the use of equivalent axis and ranges of values. Right now, both figures do not share the same range of values, and as it was displayed in Fig. 3 of Bolibar et al. (2022), there is a reduced range of values that will be encountered by French Alpine glaciers in future climate scenarios for different RCPs. This is particularly problematic for the case of winter snowfall anomalies. In Fig. 2 of this study, slight nonlinearities are displayed below -1.2 m.w.e. and above +1.7 m.w.e. These values are way beyond anything that will be seen in the 21st century for French alpine glaciers, as displayed in the vertical dashed lines in Fig. 3 of Bolibar et al. (2022). The most extreme values that French alpine glaciers will see until 2100 will range between -0.7 m.w.e. to +1.2 m.w.e. Anything beyond these limits makes no sense from a physical point of view for this sort of analyses, and will have no impact in projections for this century. At the very least, Vincent and Thibert should admit that nonlinearities of MB shown by Bolibar et al. (2022) for very extreme anomaly values out of the range of future likely encountered values, must simply not be taken into account in their analyses. Additionally, one aspect that is not mentioned in this study is the fact that they are comparing the response of two glaciers with that of 660 glaciers. Bolibar et al. (2022) reported a strong variability in terms of mass balance sensitivity response to climatic forcings along different types of glaciers. The very reduced sampling used by Vincent and Thibert shows just a partial picture of all glaciers in the region. This should be specifically mentioned when presenting the comparisons. In order to better understand these effects and to better compare both methods, I believe it is necessary to add the response of summer snowfall anomalies to Fig. 2. This would allow a comparison with the most meaningful response of the methods of Bolibar et al. (2022). Moreover, the future ranges of extreme values encountered by these glaciers under future climate scenarios (e.g. using the ADAMONT8 product which is compatible with the SAFRAN9 product used in this study), should be added to Figs. 2 and 3. This would clearly indicate where the nonlinearities actually will come into play and where they will be just model extrapolations beyond physically plausible values. This would also show that the nonlinearities linked to winter snowfall anomalies illustrated in Fig. 3 of Bolibar et al.(2022) will never occur during the 21st century in the French Alps, as they are out of the range of the values simulated by climate models.

*As suggested in the previous comment, we have done numerical experiments to analyse the summer snowfall anomalies.*
*The results are reported below:*

[Figure]

*Figure: Surface mass balance anomalies against the summer snow fall anomalies, at Argentière glacier.*

*We found an almost linear response of SMB to summer snow fall anomalies.*
*These results have been checked using the synthetic data. Here below, the mass balance anomalies have been calculated from synthetic data we have used in our paper. The snowfall anomaly has been changed by increment of +/- 100%. The sensitivity remains almost unchanged.*

[Figure]

*Figure: Surface mass balance calculated from synthetic data with different summer snow fall anomalies.*

*One can conclude that the response of SMB to summer snow fall anomalies, using degree day model is almost linear. The annual mass balance anomaly cannot be detected because the summer snow falls are low and do not affect significantly the sensitivity. Although the summer snow fall can affect the summer mass balance, the response to summer mass balance anomaly is almost linear. It is consistent with in situ observations given the low quantity of summer snow fall on alpine glaciers. In addition, in the future, the summer snowfall will be increasingly lower. It is very surprising that Bolibar et al. (2022) encountered that the strongest nonlinear response (from a statistical point of view) came from summer snowfall anomalies.*
*However, this new discrepancy with Bolibar et al. study is beyond the scope of our paper. Again, the topic of our paper is to demonstrate that the responses of degree day model to temperature and winter accumulation are non linear.*

*About the ranges of values of temperature and winter snow anomaly, the Reviewer 1 wrote: "the future ranges of extreme values encountered by these glaciers under future climate scenarios (e.g. using the ADAMONT8 product which is compatible with the SAFRAN9 product used in this study), should be added to Figs. 2 and 3."*
*This request is beyond the scope of our study, given that our unique purpose is to show that temperature-index models are able to capture nonlinear responses of glacier mass balance (MB) to high deviations in air temperature and solid precipitation.*

*The last comment of CG4 of Reviewer 1 is surprising: " Moreover, the future ranges of extreme values encountered by these glaciers under future climate scenarios (e.g. using the ADAMONT8 product which is compatible with the SAFRAN9 product used in this study), should be added to Figs. 2 and 3. This would clearly indicate where the nonlinearities actually will come into play and where they will be just model extrapolations beyond physically plausible values. This would also show that the nonlinearities linked to winter snowfall anomalies illustrated in Fig. 3 of Bolibar et al.(2022) will never occur during the 21st century in the French Alps, as they are out of the range of the values simulated by climate models. "*

*This last comment is surprising given that the nonlinearities to winter snowfall anomalies have been previously demonstrated in Réveillet et al (2018) using in situ observations and energy balance*

*modelling (see Figure 6b of Réveillet et al, 2018, replicated below in the reply to the comments of lines 97-98). It clearly shows that the nonlinearities linked to winter snowfall anomalies occurred during the 21st century in the French Alps yet.*

*Réveillet, M., Six, D., Vincent, C., Rabatel, A., Dumont, M., Lafaysse, M., Morin, S., Vionnet, V., and Litt, M.: Relative performance of empirical and physical models in assessing the seasonal and annual glacier surface mass balance of Saint-Sorlin Glacier (French Alps), The Cryosphere, 12, 1367–1386, https://doi.org/10.5194/tc-12-1367-2018, 2018.*

**1.5 GC5: Code and data availability**

Another aspect that makes it hard to understand the methods is the fact that the source code used for this study is not shared. Following the principles of open science from The Cryosphere journal, I would strongly encourage the authors to share their code and data in an open repository (e.g. GitHub). This would make the study reproducible, and it would make it easier for reviewers and readers to understand what has been done. If the authors strongly oppose to this, I would still ask them to privately share their code for this review in order to correctly understand what has been done.

*Field data are accessible through the project website at https://glacioclim.osug.fr.*
*Results from the PDD simulations on synthetic data are now accessible from the open data repository: 10.5281/zenodo.7603415.*
*It has been added in the Code and Data Avaibility of the new version of our manuscript.*

**2   Specific comments**

• L1 The current title does not give much information on what the sensitivity is linked to. I believe a correct title should be something like ''Nonlinear sensitivity of glacier mass balance to future climate change attested by temperature-index models''.

*Agree. The title has been changed: "Nonlinear sensitivity of glacier mass balance to climate attested by temperature-index models"*

• L18 The aspects regarding GC3 should be added here in the abstract.

*The abstract length cannot exceed 100 words. In addition, and as mentioned above, it is beyond the scope of our study to discuss the numerous potential causes of nonlinear response of mass balance to climate change. Our present study just aimed to prove that a simple temperature index-model is able to reproduce nonlinear responses of glacier mass balance to temperature and precipitation.*

• L33-35 This is one of the cherry picking instances mentioned in GC1. To be adjusted accordingly.

*Disagree. As explained in detail in the reply to the comment 1, our statements do not result from"cherry picking".*

• L53 How has the temperature been downscaled to be used in the temperature-index model? Two versions of SAFRAN exist: one divided by massifs and altitudinal bands, and another one in a grid. Which one of the two has been used?

*Agree. Details have been added in the data section.*

 • L61 As per the comments on GC2: how have been these two DDFs been obtained ?

*Details have been added in the Method section.*

• L79 This is one of the cherry picking instances mentioned in GC1. To be adjusted accordingly.

*Disagree. As explained in detail in the reply to the comment 1, our statements do not result from"cherry picking"*

• L97-98 This sentence is lacking solid arguments to back it. Could you please elaborate? Which parts of the model calibration might have issues? All the models in Bolibar et al. (2022) were cross-validated, ensuring a correct out-of-sample validation and a good generalization outside the seen dataset. This study so far does not provide any information regarding parameter calibration. In order to correctly compare both models and draw conclusions, a good understanding of both model calibration strategies is necessary.

*We found a sensitivity increase with low winter-accumulation anomalies using our model. The physical reasons are explained in lines 111-121.*
*These conclusions are consistent with in-situ observations (Six and Vincent, 2014).*
*It is also consistent with the results of Reveillet et al. (2018) using observations and energy balance modelling. It has been well illustrated in Figure 6b of this paper:*

[Figure]

*Figure 6b of Réveillet et al. (2018): Surface mass balance at stake 10, over one hydrological year, using averaged summer conditions (over 1996–2015), 2000–2001 winter conditions (pink) and 2008–2009 winter conditions (blue), representing the two extreme results.*

*In Réveillet et al. (2018), the tests of the annual mass balance sensitivity to seasonal mass balance using the Crocus model were performed at seven stakes in the ablation area, ranging between 2700 and 2870 m a.s.l. Only the results for stake 10 (located at 2760 m a.s.l.) are presented in Fig. 6, but conclusions are similar for all the stakes. The difference between winter accumulation of these 2 years (2001 and 2009) is 1.2 m w.e. Using the same summer conditions, the difference at the end of the hydrological year is 2.4 m w.e. (i.e. twice the difference at the end of the winter season). Thus, from these results, one can conclude that the sensitivity of annual mass balance to winter accumulation is close to 100 % when the winter accumulation is close or exceeds 2.4 m w.e. (because the glacier is covered by the snow over the whole melting season). In the contrary, with very low winter accumulation, as observed in 2008/2009, the sensitivity of annual mass balance to winter accumulation is close to 200 % (1.2 to 2.4 m w.e). These results from observations and energy balance modelling are very consistent with the results shown in Figure 3 of our paper at 2750 m (right panel): the sensitivity of annual mass balance to winter accumulation is about 110 % when the anomaly of winter accumulation exceeds 1 m w.e. On the contrary, the sensitivity of annual mass balance to winter accumulation is very close to 200 % when the anomaly of winter accumulation is lower -1 m w.e.*
*In conclusion, these results show (i) a strong non-linear effect (ii) and an obvious increase in MB sensitivity with low-winter accumulation.*

*Therefore, a temperature index model (our study), a SEB model (Reveillet et al.; 2018) and field observations (Six and Vincent, 2014) report increased sensitivities of MB under low winter-accumulation conditions. The opposite result is obtained from the deep-learning model that is why one*

*may suspect an issue in the calibration of the ANN model. Reviewer 3 also comments the ANN descrepancy on that point as "really intriguing".*

*Réveillet, M., Six, D., Vincent, C., Rabatel, A., Dumont, M., Lafaysse, M., Morin, S., Vionnet, V., and Litt, M.: Relative performance of empirical and physical models in assessing the seasonal and annual glacier surface mass balance of Saint-Sorlin Glacier (French Alps), The Cryosphere, 12, 1367–1386, https://doi.org/10.5194/tc-12-1367-2018, 2018.*

*Six, D. and C. Vincent (2014), Sensitivity of mass balance and equilibrium-line altitude to climate change in the French Alps. J. Glaciol. 60, 867–878. doi:10.3189/2014JoG14J014*

• L99-101 This is indeed true, and has already been reported in other studies. However, as argued in Bolibar et al. (2022) and as I explained in GC3, this is only part of the picture. This should be adjusted to mention that this is one of the multiple nonlinear processes in glacier mass balance sensitivity to climatic forcing, and that this process is a different one that the one reported in Bolibar et al. (2022).

*As mentioned above, it is beyond the scope of our study to discuss the numerous potential causes of nonlinear response of mass balance to climate change. Our present study just aimed to prove that a simple temperature index-model is able to reproduce nonlinear responses of glacier mass balance to temperature and precipitation.*
*In addition, the paper in Brief communication should be short.*

• L104-106 This is again a case of cherry picking. Bolibar et al. (2022) never claimed that ALL models in GlacierMIP 210 (not GlacierMIP 1, Hock et al. (2019), as stated by the authors) have linear relationships to PDDs and precipitation.

*This comment is very surprising and at complete odds with Bolibar et al. (2022) wrote (page 8), second column (lines 6-15) :*
*« Despite the existence of a wide variety of different approaches to simulate glacier dynamics, **all glacier models in GlacierMIP rely on MB models with linear relationships between PDDs and melt, and precipitation and accumulation**. Some of these models use a single DDFs, while others have separate DDFs for snow and ice, producing a piecewise function composed of two linear sub-functions that can partially account for non linear MB dynamics depending on the snow pack. **As we have previously shown, these models present a very similar behaviour to the linear statistical model from this study** ».*
*We do not understand this comment.*

L.104-106: To begin with, some of them use SEB. Bolibar et al.(2022) make a point that temperature-index models with a single DDF clearly behave like the Lasso; and even temperature-index models with 2 DDFs, can only partially account for the nonlinearities (and cannot capture the ones they show in their study). This is further corroborated by the comparisons made in that study between the Lasso MB model and the temperature-index MB model from GloGEM in the Supplementary material of Bolibar et al. (2022). To be modified accordingly.

*The authors Bolibar et al. (2022) are faced with a paradox: 1) they have to demonstrate that their ANN model captures a non linear response of glaciers to climate which has not be seen in the previous models, as claimed in their Title, but 2) they cannot deny that temperature-index models account for nonlinearities as recognized here and in our study.*

• L107-108 This is not accurate. The Open Global Glacier Model (OGGM), which was used in the paper as an example of this behaviour, also has a single DDF. In order to avoid further cherry picking, I would ask the authors to be precise about their claims. Out of the of the 11 models in Marzeion et al. (2020)10, 7 are using temperature index models (2 of them with a single DDF), 1 is using a simple parametrizations relating MB to air temperature, 1 is using a mass balance gradient based on temperature indices, and 2 are using surface energy balance models. This means that at least 3

(potentially 4 if we take into account Kraaijenbrink et al. (2017)) models have direct simple linear relationships between PDDs and MB. The other 4 have 2 DDFs, which can partially account for nonlinearities (but not the ones from the above-mentioned point 1 in the temporal dimension).

*Here, we refer to **temperature-index models** of the Glacier Model Intercomparison Project (GlacierMIP) (Hock et al., 2019). Except for one model (that of Marzeion et al., 2014), all **temperature-index models** used in GlacierMIP include two degree-day factors and account for nonlinearities.*
*In Marzeion et al. (2020), 5 **temperature-index models** out of 7 **temperature-index models** used in GlacierMIP include two degree-day factors.*
*Anyway, we changed our statement and replaced: "This is erroneous given that, except for one model (that of Marzeion et al., 2014), all temperature-index models used in GlacierMIP include two degree-day factors." by "This is erroneous given that, most of the temperature-index models used in GlacierMIP include two degree-day factors."*

• L117-119 This is one of the cherry picking instances mentioned in GC1. To be adjusted accordingly.

*Disagree. As explained in detail in the reply to the comment 1, our statements do not result from"cherry picking"*

• L124-126 This was already done in that study. The results were shown in the Supplementary material. The exact same plots were not produced due to the difficulty of implementing that scheme on GloGEM. But the evolution of the MB for future scenarios was compared, yielding very similar results and responses to those of the LASSO. Therefore, the comparison between the LASSO and the TI model from GloGEM in terms of projected cumulative MB is not unfounded. Vincent and Thibert must point this aspect in an objective manner.

*We do not understand how the sensitivities can be similar between a TI model and LASSO model. Anyway, the sentence "We would suggest testing the capability of an ANN to capture nonlinearity by comparing its results with that of the GloGEM Positive Degree-Day (PDD) model that they used in their paper." has been removed from our manuscript. It is also a suggestion of Reviewer 3.*

• L129-131 As previously discussed in GC3, this is because the TI model used in this study does not account for DDF evolution over time. To be mentioned here in order to clarify the bigger picture.

*We disagree. If the DDF were to change from year to year to account for the darkening of the ice due to longer ice exposition to the atmosphere (dust deposition, etc…), the sensitivity of MB to winter balance would much more increase under low winter accumulation conditions. Our model effectively does not account for such a process. Accounting for would increase the discrepancy with the ANN approach at this step.*
*In any case, it is beyond the scope of our analyses.*

---

## Author Comment (AC2)

**Reply to Reviewers :**

*We have complied with all the comments made by Reviewers (see the point-by-point response). The replies are in italics.*

*The manuscript has been revised by a professional native speaker (not this Reply).*

**Reply to Reviewer 2**

 **General comments** :

The submitted manuscript presents numerical experiments on glacier surface mass balance (SMB) based on a temperature index (TI) model. The authors aim on demonstrating that even with linear relationships between air temperature and snow/ice melt, such models show a non-linear sensitivity to climatic variations. The manuscript is a direct response to a publication by Bolibar et al. (2022), which presents comparisons between a deep ANN approach for estimating the glacier response to climate projections with simple TI models.

Vincent and Thibert criticise the proposition in Bolibar et al. (2022) that simple TI models do not show a non-linear sensitivity to climate variations in their experiments, in contrast to the deep ANN experiments. This is not the place to discuss details and validity of the Bolibar et al. (2022) paper, but rather to review the issues presented in the manuscript at hand. The authors raise an interesting question, which was also discussed earlier: what is the characteristic behaviour of TI models for changing climatic boundary conditions? It seems that one potential conflict is not as severe as it is presented. Even though Bolibar et al. (2022) state that TI models usually show a linear response to climate variations, they mainly compare a fully linear Lasso approach with their ANN model. They even claim that TI models with different degree day factors for snow and ice melt, show some non-linear behaviour, but that their response is not adequate compared to the ANN approach. Therefore, the response of Vincent and Thibert should be focussed on the validity of the TI model response, rather than on just demonstrating the non-linearity per se.

However, this manuscript provides rather interesting insights into the fundamental behaviour of TI models and this analysis could serve as a great study about the model characteristics, if some shortcomings could be fixed. The authors concentrate on demonstrating the non-linear response on a change in forcing, instead of discussing the fundamental interaction of the differences in snow and ice melt for the final glacier mass balance. There is a multitude of publications, which discuss the limitations of TI models due to their fixed relationship between air temperature and melt, while temperature is an indicator of energy availability, not energy transfer. But as long as the forcing stays within certain limits, TI models provide a robust and simple method for SMB estimates. Therefore, the critical investigation of the non-linearity characteristics within these limits would add high value to the discussion, also in the light of the application of AI approaches.

*Thanks for your comments. We agree that this is not the place to discuss details and validity of the Bolibar et al. (2022) paper. The purpose of our study is to discuss the glacier response to climate projections with simple TI models only.*
*Given that temperature-index models have been widely used by the glaciological community for glacier projections in large-scale studies over the 21$^{st}$ century, we believe that it is crucial to clarify these issues.*

**Major concerns**:

The data section does not provide the necessary information to evaluate the experiments. Only the two glaciers are described, but details neither about the mass balance data are given, nor about the necessary additions information, like DEMs etc. The methods section does not provide sufficient details. It is not clear how the model is applied to the glaciers. Is it a spatially distributed model, which cell resolution is used, is the glacier surface elevation static, or is there a dynamic response? What is the time step? How was the forcing parameterised across the elevations/aspects?

With regard to the general criticism of the Bolibar et al. (2022) paper, it needs to be highlighted that they estimate SMB for an entire region, while here two individual glaciers are considered. This allows a more detailed investigation of local SMB reaction, compared to general trends. There is no section about the determination of the DDF values. I would expect a section about calibration and validation with the available forcing data set, or at least information where to find these details.

*Details about mass balance and DEM data are now given in the Data section.*
*The point mass balances are calculated for each elevation, for Argentière and Sarennes glaciers. In addition, we calculated the glacier-wide mass balance of Argentière glacier using the point mass balances for elevation range and geodetic mass balances (Vincent et al., 2009).*
*The degree-day factors for snow and ice are 0.0035 and 0.0055 m w.e. $K^{-1}d^{-1}$ for Argentière glacier (Reveillet et al., 2017) and 0.0041 and 0.0068 m w.e. $K^{-1}d^{-1}$ for Sarennes glacier (Thibert et al., 2013). The calibration and validation of these factors have been done in (Reveillet et al., 2017) and (Thibert et al., 2013). It is not possible to provide more details in the present paper but further information can be found in these papers.*

In the Results section, you describe the non-linearity of the model response with an increase of sensitivity with respect to the anomaly (L.81). A major point would be to relate magnitude of these sensitivities to the sensitivities found by Bolibar et al. (2022) with their deep ANN approach and discuss the consequences within the bounds of potential future anomaly ranges.

*It is not easy to compare the magnitude of sensitivities obtained in our study and found by Bolibar et al. (2022). Indeed, in the paper of Bolibar et al. (2022), the anomalies are calculated and averaged (i) from 660 glaciers, (ii) from glacier-wide mass blances of these glaciers, (iii) over the 1967-2015 period.*
*The purpose of our study is to show that responses in MB are not linear to temperature or precipitation changes even using a simple degree-day model. We used the point mass balances because they are free from dynamics impact (surface changes). The surface changes which influence the glacier-wide mass balance can lead to additional non-linearities. But here, we aim to discuss the response of mass balance using TI model only.*
*As seen in Figure 3, the response of annual mass balance to air temperature and to winter accumulation is different according to the elevation. The magnitude of these sensitivities are also different. It is also different from a glacier to another one. Thus, it is difficult to relate magnitude of these sensitivities to the sensitivities found by Bolibar et al. (2022) with their deep ANN approach using glacier-wide mass blances of 660 glaciers. It would require to apply the degree-day model on 660 glaciers, which is beyond the scope of our study.*
*Our topic is to show that the responses in MB calculated with a simple degree-day model are not linear to temperature or precipitation changes. Except the very high elevations where the glacier is covered by snow during the whole year, we demonstrate that the response of calculated annual mass balance using TI model to meteorological variables is not linear.*

I wonder why you did not investigate summer snow fall in your experiments, as this is the major actor of non-linear response in the ANN approach. It should also be expected that summer snow fall has a strong non-linear response in the TI model, because of the difference in DDF values for snow and ice and the strong reduction of melt in the main ablation season.

*According to this suugestion, we have done numerical experiments to analyse the summer snowfall anomalies.*
*The results are reported below:*

[Figure]

*Figure: Surface mass balance anomalies against the summer snow fall anomalies, at Argentière glacier*

*We found an almost linear response of SMB to summer snow fall anomalies. These results have been checked using the synthetic data. Here below, the mass balance anomalies have been calculated from synthetic data we have used in our paper. The snowfall anomaly has been changed by increment of +/- 100%. The sensitivity remains almost unchanged.*

[Figure]

*Figure: Surface mass balance calculated from synthetic data with different summer snow fall anomalies.*

*One can conclude that the response of SMB to summer snow fall anomalies, using degree day model is almost linear. The annual mass balance anomaly cannot be detected, probablybecause the summer snow falls are low and do not affect significantly the sensitivity. Although the summer snow fall can affect the summer mass balance, the response to summer mass balance anomaly is almost linear. It is consistent with in situ observations given the low quantity of summer snow fall on alpine glaciers. In addition, in the future, the summer snowfall will be increasingly lower. It is very surprising that Bolibar et al. (2022) encountered that the strongest nonlinear response (from a statistical point of view) came from summer snowfall anomalies.*
*However, this new discrepancy with Bolibar et al. study is beyond the scope of our paper given that the topic of our paper is to demonstrate that the responses of degree day model to temperature and winter accumulation are non linear.*
*Given the limited format of "Brief communication", we did not add anything about the response of annual mass balance to the summer snow fall anomalies.*

Sensitivity to winter balance: in L.94-98 you describe the contrasting results of the TI model with respect to the ANN model with regards to a decreasing winter balance. However, there needs to be an explanation why the TI model explains reality and what are the consequences in the view of the ANN results. As ANN is more or less a black box, the results cannot be judged in the view of physical constraints, but just in respect to validation data sets. An investigation on the physical basis for the TI models' sensitivity would improve the discussion about the pros and cons of the two different approaches.

*To explain the increased sensitivity of mass balance with decreasing winter accumulation, we performed runs of our PDD model on synthetic data under different conditions of winter balance. The results are reported in Figure 5.*
*The results show that the increase in sensitivity can be physically explained by the earlier disappearance of the winter snow cover. The earlier increase in the ablation rate under lower conditions of winter balance results in nonlinearity attested by the spread between MB plots in Figure 5b. For instance, with winter accumulation decreased by -1500 mm, the ice ablation starts very early (by the end of May) and the annual MB is close to -5.55 m w.e. $a^{-1}$ in October. With winter*

*accumulation increased by +1500 mm, the ice ablation starts in mid-September and the annual MB is close to -0.21 m w.e. a$^{-1}$ in October.*

*Our findings are consistent with in-situ observations (Six and Vincent, 2014).*

*It is also consistent with the results of Reveillet et al. (2018) using observations and energy balance modelling. It has been well illustrated in Figure 6b of this paper:*

[Figure]

*Figure 6b of Réveillet et al. (2018 Surface mass balance at stake 10 (2760 m a.s.l.), over one hydrological year, using averaged summer conditions (over 1996–2015), 2000–2001 winter conditions (pink) and 2008–2009 winter conditions (blue), representing the two extreme results.*

*In Reveillet et al. (2018), the tests of the annual mass balance sensitivity to seasonal mass balance using the Crocus model were performed at seven stakes in the ablation area, ranging between 2700 and 2870 m a.s.l. Only the results for stake 10 (located at 2760 m a.s.l.) are presented in Fig. 6b above, but conclusions are similar for all the stakes. The difference between winter accumulation of these 2 years (2001 and 2009) is 1.2 m w.e (Fig. 6b). Using the same summer conditions, the difference at the end of the hydrological year is 2.4 m w.e. (i.e. twice the difference at the end of the winter season). Thus, from these results, one can conclude that the sensitivity of annual mass balance to winter accumulation is close to 100 % when the winter accumulation is close or exceeds 2.4 m w.e. (because the glacier is covered by the snow over the whole melting season). In the contrary, with very low winter accumulation, as observed in 2008/2009, the sensitivity of annual mass balance to winter accumulation is close to 200 % (1.2 to 2.4 m w.e). These results from observations and energy balance modelling are very consistent with the results shown in Figure 3 of our paper at 2750 m (right panel): the sensitivity of annual mass balance to winter accumulation is about 110 % when the anomaly of winter accumulation exceeds 1 m w.e. On the contrary, the sensitivity of annual mass balance to winter accumulation is very close to 200 % when the anomaly of winter accumulation is lower -1 m w.e.*
*One can conclude that these results show (i) a strong non-linear effect (ii) and an obvious increase in MB sensitivity with low-winter accumulation.*
*Some explanations have been added in the manuscript.*

*Réveillet, M., Six, D., Vincent, C., Rabatel, A., Dumont, M., Lafaysse, M., Morin, S., Vionnet, V., and Litt, M.: Relative performance of empirical and physical models in assessing the seasonal and annual glacier surface mass balance of Saint-Sorlin Glacier (French Alps), The Cryosphere, 12, 1367–1386, https://doi.org/10.5194/tc-12-1367-2018, 2018.*

*Six, D. and C. Vincent (2014), Sensitivity of mass balance and equilibrium-line altitude to climate change in the French Alps. J. Glaciol. 60, 867–878. doi:10.3189/2014JoG14J014*

**Minor issues:**

L.24: I would prefer "Surface mass balance" projections instead of "glacier mass projections", as the projections aim on SMB not on the full mass variations (which include basal melt and other processes).

*It has been changed.*

L.24-29: There should be at least a short characterisation of the differences in model approaches, in order to clarify the topic.

*The temperature index model is described in Method Section. We added some explanations about the ANN approach: « A neural network is a collection of interconnected simple processing elements called neurons. These processing elements are connected with coefficients or weights, which constitute the neural network structure. Every connection of a neural network is assigned a weight that comes through training the ANN (Agatonovic-Kustrin and Beresford, 2000). »*

*Agatonovic-Kustrin, S. and Beresford, R. : Basic concepts of artificial neural network (ANN) modeling and its application in pharmaceutical research, Journal of Pharmaceutical and Biomedical Analysis, 22, 5, https:// doi.org./ 10.1016/s0731-7085(99)00272-1,2000.*

L.37-38: Already here it would be helpful to shortly discuss the basic non-linearity of coupled linear relationships.

*In Introduction, we present the questions and the topic of our paper. We do not think more explanations about non-linearity are required here in Introduction. It is thoroughly discussed later in the manuscript*

L.54: The information about the reanalysis data needs to be described in the data section. It requires also some information on periods used, resolution etc.

*It has been done.*

L.64: Is k a function or just a two-value parameter?

*Parameter k is depending on the site elevation to account for the precipitation gradient and is determined from winter balance measurements and precipitation data. Some information have been added in Method section.*

L.74: how was the anomaly applied to the original data? I guess that you applied a constant anomaly to the daily values of the forcing series, in order to calculate a SMB anomaly.

*The anomaly is a shift of the mean of the distribution of the original data in temperatures and winter balances. We kept the same distribution around the means to reproduce the year-to-year variability.*
*We added the following sentence in the manuscript. "The anomaly is generated as a shift (increment/decrement) of the mean of the distribution of the original data in temperatures and winter balances. The distribution around the means is unchanged (same year-to-year variability as found in the original data)."*

L.77: It is not clear what you did here. I assume that you ran the model at specific points (where you presumably also have stake information) and as a distributed model across the entire glacier (which grid, etc.?).

*We calculated the response of point mass balance at 2750 m and 3100 m on the Argentière and Sarennes glaciers (Fig. 2). In addition, we calculated the response of point mass balance at 2450 m, 2750 m, 3250 m a.s.l. and the response of glacier-wide mass balance on the Argentière glacier (Fig. 3). This information has been added in the manuscript.*

L.79: Your statement with respect to Bolibar et al. (2022) is not exactly correct: Bolibar et al. (2022) write about piecewise linear relationships and implied non-linear response on page 5. However, their conclusion is that the TI model results are rather similar to the Lasso approach, which is clearly not confirmed by your investigations.

*We agree that Bolibar et al. (2022) write about piecewise linear relationships and implied non-linear response on page 5. Indeed, in the Results section of Bolibar et al. (2022), on can read : « In that study, a temperature-index model with a separate degree-day factor (DDF) for snow and ice is used, resulting in piecewise linear functions able to partially reproduce nonlinear MB dynamics ».*
*However, they write in the following sentences: « Both the Lasso and the temperature-index MB model rely on linear relationships between PDDs, solid precipitation and MB. Therefore, their sensitivities to the projected 21st century increase in PDDs are linear. »*

*In addition, the statement is clear in the abstract of Bolibar et al. (2022), one can read : « Deep learning captures a nonlinear response of glaciers to air temperature and precipitation, improving the representation of extreme mass balance rates compared to linear statistical and temperature-index models. »*

*These statement is repeated at numerous places in the manuscript.*

*In the Introduction of Bolibar et al. (2022), on can read : « This type of model uses a calibrated linear relationship between positive degree-days (PDDs) and the melt of ice or snow. The main reason for their success comes from their suitability to large-scale studies with a low density of observations, in some cases displaying an even better performance than more complex models. However, both the climate and glacier systems are known to react non-linearly, even to pre-processed forcings like PDDs, implying that these models can only offer a linearized approximation of climate-glacier relationships. »*

*In the results section of Bolibar et al. (2022), on can read : « In that study, a temperature-index model with a separate degree-day factor (DDF) for snow and ice is used, resulting in piecewise linear functions able to partially reproduce nonlinear MB dynamics.*

*In Discussion of Bolibar et al. (2022), the authors justify tha analogy between Lasso model and Temperature index model. They write : « At this point, it is important to clarify the different ways of treating PDDs in the Lasso and the temperature-index MB models analysed in this study in order to justify analogies » and they conclude : « Nonetheless, since they are both linear, their calibrated parameters establishing the sensitivity of melt and glacier-wide MB to temperature variations remain constant over time. »*

*When they analyse the glacier models in GlacierMIP, they bring some nuances : « Despite the existence of a wide variety of different approaches to simulate glacier dynamics, all glacier models in GlacierMIP rely on MB models with linear relationships between PDDs and melt, and precipitation and accumulation. Some of these models use a single DDF, while others have separate DDFs for snow and ice, producing a piecewise function composed of two linear sub-functions that can partially account for nonlinear MB dynamics depending on the snowpack", but they finally conclude that « As we have previously shown, these models present a very similar behaviour to the linear statistical MB model from this study »*

*Finally, in the last paragraph of Bolibar et al. (2022), the authors claim that « By unravelling nonlinear relationships between climate and glacier MB, we have demonstrated the limitations of linear statistical MB models to represent extreme MB rates in long-term projections. Our analyses suggest that these limitations can also be translated to temperature-index MB models, as they share linear relationships between PDDs and melt, as well as precipitation and accumulation ».*

*Thus, we believe that Bolibar et al. (2022) clearly question the ability of temperature-index models to capture nonlinear responses of glacier surface-mass balance (SMB) to high deviations in air temperature and solid precipitation*

L.82: Details about the synthetic input series are missing (how did you construct these series? Do they represent a certain realistic SMB range?).

*We added the following sentence in the manuscript:*
*"The reference scenario (unforced temperature and winter balance reference conditions) of synthetic data is typical for a location in the upper ablation area of an Alpine glacier. "*

*Further information are given in the following sentences:*

*"Runs of our PDD model on synthetic data under different conditions of winter balance (Fig. 5) used a reference scenario of 1,700 mm of winter balance changed by increments of ±300 mm in precipitation". Temperature data are shown in Figure 4a, typical for a course of atmospheric temperature from spring to autumn around 3000 m of elevation in the Alps. We use PDD factors for snow and ice from Thibert et al. (2013).*
*In addition, results from the PDD simulations on synthetic data are now accessible from the open data repository:* 10.5281/zenodo.7603415.
*It has been added in the Code and Data Avaibility of the new version of our manuscript.*

L.85: It might be a good idea to show the length of ice ablation period vs total ablation period and the onset date of ice ablation. The onset of ice ablation is a measure of the nonlinear character.

*According to your suggestion, we have done the following Figure:*

[Figure]

*It shows the ice ablation duration with respect to the total ablation duration for different temperature forcing. It shows the non linear response. However, we did not include this Figure in the manuscript in order to avoid to lengthen the manuscript which is a Brief Communication".*

L.91: You describe an increase in sensitivity, but this should be quantified with respect to the disappearance of winter snow to judge the physical basis.

*To explain the increased sensitivity of mass balance with decreasing winter accumulation, we performed runs of our PDD model on synthetic data under different conditions of winter balance. The results are reported in Figure 5.*

*The results show that the increase in sensitivity can be physically explained by the earlier disappearance of the winter snow cover. The earlier increase in the ablation rate under lower conditions of winter balance results in nonlinearity attested by the spread between MB plots in Figure 5b. For instance, with winter accumulation decreased by -1500 mm, the ice ablation starts very early (by the end of May) and the annual MB is -5.55 m w.e. a-1 in October. With winter accumulation increased by +1500 mm, the ice ablation starts in mid-September and the annual MB is -0.21 m w.e. a-1 in October. This asymmetry clearly shows that the response to winter accumulation is not linear.*

L.110-111: There is a basic difference between the Lasso-models and the TI approach, as the second one uses a step function of the DDF parameter. Therefore, it cannot be expected that the two models provide the same response. This should be made clear.

*The LASSO MB model is based on a regularized multi-linear regression. We added some explanations in the manuscript.*

*At numerous places in the manuscript of Bolibar et al. (2022), the authors claim that Lasso model and Temperature-index model both provide linear responses. In Results section, they wrote: « Both the Lasso and the temperature-index MB model rely on linear relationships between PDDs, solid precipitation and MB. Therefore, their sensitivities to the projected 21st century increase in PDDs are linear. »*
*In Discussion, the authors justify the analogy between Lasso model and Temperature index model. They write : « At this point, it is important to clarify the different ways of treating PDDs in the Lasso and the temperature-index MB models analysed in this study in order to justify analogies » and they conclude : « Nonetheless, since they are both linear, their calibrated parameters establishing the sensitivity of melt and glacier-wide MB to temperature variations remain constant over time. »*

*However, the discussion about the differences between LASSO model and temperature index model is beyond the scope of our study, which is focused on the non-linearity of Temperature Index models.*

L.117: It might be a good idea to mention also earlier investigations who pointed out this basic behaviour.

*As mentioned in the reply to general comments, our findings are consistent with in-situ observations (Six and Vincent, 2014). It is also in agreement with the results of Reveillet et al. (2018) using observations and energy balance modelling. Some explanations have been added in the manuscript.*

*Réveillet, M., Six, D., Vincent, C., Rabatel, A., Dumont, M., Lafaysse, M., Morin, S., Vionnet, V., and Litt, M.: Relative performance of empirical and physical models in assessing the seasonal and annual glacier surface mass balance of Saint-Sorlin Glacier (French Alps), The Cryosphere, 12, 1367–1386, https://doi.org/10.5194/tc-12-1367-2018, 2018.*

*Six, D. and C. Vincent (2014), Sensitivity of mass balance and equilibrium-line altitude to climate change in the French Alps. J. Glaciol. 60, 867–878. doi:10.3189/2014JoG14J014*

L.121-126: This is a solid argumentation and provides a core conclusion. But is should be expanded by the major points, mentioned above.

*Thank you for your comment.*
*We do not believe that we can expand our manuscript given that our paper is a Brief Communication.*

L.129: It is only mentioned that the physical reasons are given for a higher sensitivity of TI models to lower winter MB, but I did not find a sound discussion.

*It has been discussed in details earlier in the manuscript:*

*"Concerning the winter balance, we found a nonlinear response of MBs to winter precipitation with our PDD model and this is also inconsistent with the conclusions of Bolibar et al. (2022) relative to the sensitivity of temperature-index models. Runs of our PDD model on synthetic data under different conditions of winter balance (Fig. 5) used a reference scenario of 1,700 mm of winter balance changed by increments of ±300 mm in precipitation. Results show that the increase in sensitivity can be physically explained by the earlier disappearance of the winter snow cover. The earlier and abrupt increase in the ablation rate under lower conditions of winter balance (Fig.5a) results in nonlinearity attested by the spread between MB plots in Figure 5b. Surprisingly, we detect sensitivity to winter accumulation, contrary to the Bolibar et al. (2022) findings using their ANN (Fig. 2 and 3). Indeed, MB sensitivity increases with low winter-accumulation anomalies using our model, but decreases in the deep-learning model of Bolibar et al. (2022). The opposite results obtained from the deep-learning model are paradoxical and may be due to an issue in the calibration of the model."*

*Here, in the Conclusions, we summed up these results and this discussion.*

L.132-136: The main concern is, that TI models applied far outside the calibration range of parameters might not be able to represent the energy exchange between the atmosphere and snow/ice correctly. However, this might also be true for the ANN approach, because it is unclear how good the performance is far beyond the training domain. Therefore, only a comparison of the different models in a large parameter space with a physical energy balance model would provide serious assessment of the model performance. This is out of scope of this manuscript, but could be mentioned as a valuable future step.

*We agree. However, as you said, it is beyond the scope of our manuscript which is a Brief Communication with limited space.*

Data availability requires at least the references to the data sets.

*The Data are accessible from the Observatory of French glaciers through the project website at https://glacioclim.osug.fr. Results from the PDD simulations on synthetic data are now accessible from the open data repository: 10.5281/zenodo.7603415.*
*It has been added in the Code and Data Avaibility.*

---

## Author Comment (AC3)

**Reply to Reviewers :**

*We have complied with all the comments made by Reviewers (see the point-by-point response). The replies are in italics.*

*The manuscript has been revised by a professional native speaker (not this Reply).*

**Reply to Reviewer 3**

**General comments** :

This brief communication investigates the nonlinear sensitivity of simple degree-day models to changes in air temperature and precipitation. The paper is written as a direct response to the article published by Bolibar et al. (2022), Nature Communications, in order to take up one important element of that study and provide additional insights. Overall, this scientific debate is interesting, and I'm convinced that the glaciological literature will benefit from this short paper by Vincent and Thibert that makes a clear and straight-forward statement. Attesting a nonlinear sensitivity to degree-day models is not at all new both in my understanding and that of the authors, as well as J. Bolibar, Reviewer 1 to the present study, and author of the paper addressed in this brief communication. Nevertheless, I find the simplicity of the approaches and the clarity of the statement provided here very useful.
A review of the present brief communication is however not easy as we also need to appropriately consider how the statements relate to the original paper by J. Bolibar, as well as his detailed review. Therefore, the situation is relatively complex, and it will heavily rely on the editor's judgment how to weight the different arguments.
In general, I personally agree with most of the basic statements made in the present study and, hence, would recommend it for publication. However, there is still ample room for improvement in the description of the data, the methods and the presentation (see below), which can definitely be achieved by the authors with relatively limited effort. With regards to the direct opposition to the study by Bolibar – that provides a variety of valuable new insights into processes and methodologies – I would suggest to try and make several of the statements not sound as reproach but more to position this study as a stand-alone research resulting in an important and clearly presented outcome.
This would also somewhat lift the obvious conflict between the original study by J. Bolibar and this paper. Nevertheless, I should mention that I agree that putting the statements of Bolibar et al. (2022) into context is justified, and is not just "cherry picking" as it is termed by Reviewer 1: The Abstract of Bolibar et al. (2022) states twice very clearly that temperature-index models have a linear sensitivity. And even the title ("nonlinear sensitivity … unveiled …") indirectly implies that previous approaches were linear in comparison to the new model. Even though the text indeed provides additional statements that actually better agree with the outcomes of this study, it is the Title and the Abstract that defines what readers take with them. Therefore, I agree that the present paper by Vincent and Thibert, and also some of the formulations (see more details below) are justified.

*Many thanks for your comments. We agree to try and make several of our statements not sound as reproach. For instance, in Introduction, the sentence "Their paper questions the use of temperature-index models for projections of glacier-mass changes in response to global warming." has been removed. Our unique purpose is to show that temperature-index models are able to capture nonlinear responses of glacier mass balance (MB) to high deviations in air temperature and solid precipitation.*

**Specific comments**:

Line 14: Please reformulate to make this less sound as an opposition to Bolibar et al. (2022). In fact, I would not say that the study has "questioned" the nonlinearity in degree-day models but has maybe

not "adequately considered/presented" it in the analysis by using a LASSO model that is linear by definition.

*It is very difficult to reformulate and refer to LASSO model in the Abstract, given that the Abstract should not exceed 100 words. We believe it is not possible to mention LASSO model and provide further details in the Abstract. More explanations have been added in the manuscript (l. 147-149): "....In the Bolibar et al. (2022) paper, the MB anomalies in response to climate forcing were obtained using a linear LASSO SMB model, which is based on a regularized multi-linear regression.*

Line 22: Either define MB at first instance, or write out always. The use of surface mass balance (SMB) would probably be more appropriate.

*It has been changed.*

Line 27: It would be good to also explicitly refer to energy-balance models. Whereas both degree-day models and ANNs do not fully resolve the actual processes and thus heavily depend on the available calibration data, energy-balance models try to fully describe the processes and the feedbacks which is certainly the optimal approach (although yet mostly inapplicable at large scales).

*In the Introduction, we wrote "Most glacier-mass projections in response to climate change in large-scale studies spanning the 21$^{st}$ century have been based on temperature-index models (Huss and Hock, 2015; Fox-Kemper et al., 2021), given the lack of available or reliable information on detailed future meteorological variables (Réveillet et al., 2018)." We refer here to energy-balance models (see Réveillet et al., 2018).*

*Réveillet, M., Six, D., Vincent, C., Rabatel, A., Dumont, M., Lafaysse, M., Morin, S., Vionnet, V., and Litt, M.: Relative performance of empirical and physical models in assessing the seasonal and annual glacier surface mass balance of Saint-Sorlin Glacier (French Alps), The Cryosphere, 12, 1367–1386, https://doi.org/10.5194/tc-12-1367-2018, 2018.*

Line 36: Avoid the term "question" and formulate in a more neutral way. Overall, I suggest to not put the opposition to the paper by Bolibar et al. (2022) as the main motivation for the paper, but rather to focus on the research question and the statement (nonlinear sensitivity of simple degree-day models).

*Agree. The sentence "Their paper questions the use of temperature-index models for projections of glacier-mass changes in response to global warming." has been removed.*

Line 48: Define which field observations have been used. Point mass balances, seasonal, monthly? Glacier-wide mass balances? Geodetic ice volume changes?

*Details about mass balance and DEM data are now given in the Data section. The point mass balances are calculated for each elevation, for Argentière and Sarennes glaciers. In addition, we calculated the glacier-wide mass balance of Argentière glacier using the point mass balances for elevation range and geodetic mass balances (Vincent et al., 2009).*

Line 55: Same statement and formulation as on line 41

*Agree.*

Line 58: Quite some unclarity remains regarding the application of the main equation: (1) Is the equation applied for each day individually, or for the entire year just once with total precipitation / cumulative degree days (I assume the first)

*Agree. It has been clarified.*

. (2) How is it decided which DDF is being used?

*The model is able to calculate the amount of snow on the glacier (Reveillet et al., 2017)*

 (3) How are the values of the DDFs determined ? Are they the same for both glaciers investigated?

*The origin of the values of the DDFS has been explained in the new version of the manuscript. The degree-day factors for snow and ice are 0.0035 and 0.0055 m w.e. $K^{-1}d^{-1}$ for Argentière glacier (Reveillet et al., 2017) and 0.0041 and 0.0068 m w.e. $K^{-1}d^{-1}$ for Sarennes glacier (Thibert et al., 2013). The point mass balances are calculated for each elevation, for Argentière and Sarennes glaciers. In addition, we calculated the glacier-wide mass balance of Argentière glacier using the point mass balances for elevation range and geodetic mass balances (Vincent et al., 2009). Parameter k depends on the site elevation to account for the precipitation gradient and is determined from winter balance measurements and precipitation data*

(4) How is the model spatially discretized? In elevation bands, on a grid?

*It is now explained in the new version of the manuscript*

(5) How is temperature and precipitation extrapolated over the different elevations of the glacier ?

*The temperature and precipitation are obtained from SAFRAN reanalyses (Durand et al., 2009; Verfaillie et al., 2018) as explained in the Method section. Parameter k used for accumulation depends on the site elevation to account for the precipitation gradient and is determined from winter balance measurements and precipitation data*

(6) Wouldn't it make physical sense to set the threshold between solid and liquid precipitation slightly above 0 deg C? In fact, in almost all situations, snow has not transitioned into rain exactly at the melting point.

*Yes, it could make physical sense to set the threshold at a value different from the melting point. Many numerical experiments have been done in (Reveillet et al., 2017) and (Thibert et al., 2013). It is important to simulate the surface mass balance. However, the impact on the sensitivity of MB to meteorological variable is neglictible.*

Line 71: Well, probably the agreement is good because the model has been calibrated accordingly. More details on the cal-val procedure and the performance of the model (including RMSE, bias with observations) is needed.

*In the new version of the manuscript, we added some information: "The degree-day factors for snow and ice are 0.0035 and 0.0055 m w.e. $K^{-1}d^{-1}$ for Argentière glacier (Reveillet et al., 2017) and 0.0041 and 0.0068 m w.e. $K^{-1}d^{-1}$ for Sarennes glacier (Thibert et al., 2013). The calibration and validation of these factors have been done in (Reveillet et al., 2017) and (Thibert et al., 2013)."*
*It is not possible to provide more details in the present paper (Brief Communication) but further information can be found in these papers. For instance, for Sarennes modelling, Pearson R=0.93, RMS=0.44 m w.e., average deviation = 0.36 m w.e. and bias with observations: -3.3 cm w.e. (model minus data)*

Line 71: "using THESE data" – which data are you referring to here?

*We replaced the sentence in the new version: "Using these reconstructed MB…"*

Line 73: Why not the median elevation? E.g. for Sarennes the elevation chosen in likely above the median.

*As seen in Figure 3, the calculations have been done also at 3250 m a.s.l., 2750 m, 2450 m and over the entire surface of the glacier.*

Line 75: The approach of the T and P anomalies needs to be better described. So, the anomaly is the same for every day of the year?

*The anomaly is a shift of the mean of the distribution of the original data in temperatures and winter balances. We kept the same distribution around the means to reproduce the year-to-year variability.*

*A new sentence has been added in the new version of the manuscript : « The anomaly is generated as a shift (increment/decrement) of the mean of the distribution of the original data in temperatures and winter balances. The distribution around the means is unchanged (same year-to-year variability).”*

Line 82: The use of synthetic temperature data comes very abruptly. It has not been introduced in the methods. How is this synthetic series constructed, i.e. what is it based on. One problem with degree-day models that might be mentioned here or in the discussion is that calibrated parameters are often related to the characteristics of the series used. I.e. if shifting to a synthetic series, this might result in an invalidity of parameters (this might also be the case for ANN approaches). In any case, this would not question the sensitivity tests performed here but transferring the result back to real conditions is not straight-forward.

*We added the following sentence in the new version of the manuscript: “The reference scenario (unforced temperature and winter balance reference conditions) of synthetic data is typical for a location in the upper ablation area of an Alpine glacier. “*
*Further information are given in the following sentence: “Runs of our PDD model on synthetic data under different conditions of winter balance (Fig. 5) used a reference scenario of 1,700 mm of winter balance changed by increments of ±300 mm in precipitation”. Temperature data are shown in Figure 4a, typical for a course of atmospheric temperature from spring to autumn around 3000 m of elevation in the Alps. We use PDD factors for snow and ice from Thibert et al. (2013).*

Line 97: Indeed, this is a very interesting finding, and it is important to state here.

*Right.*

Line 111: Also here, I agree – this question in really justified.

*Right.*

Line 117: The wording "refute" is too strong in my opinion. This study adds an important precision / an emphasis on one aspect of the study by Bolibar et al (2022) but it does not refute the findings of that study in general.

*We changed the wording: "These results question those of Bolibar et al. (2022), which argue that temperature-index models provide only linear relationships between positive degree-days (PDDs), solid precipitation and SMB."*

Line 125: Even though I agree with this statement (see above) I find it somewhat inappropriate to ask this in the conclusion of this (formally) fully independent paper, and would thus rather omit it, or strongly reformulate. The similarity to the figure by Bolibar et al (2022) and the ANN approach presented there is really intriguing! However, I suggest making this figure and the presentation of these results (that are crucial to the study) more consistent: Please use the same ranges of the values (both x and y-axis) for both glaciers. How were these ranges determined ? Wouldn't it make sense to test exactly the same ranges as Bolibar et al (2022) ? In addition, I do not fully understand why the

authors decided to only display results for a selected elevation while results for the entire glacier (see Fig. 3) would be available. This should be better motivated.

*. The sentence "We would suggest testing the capability of an ANN to capture nonlinearity by comparing its results with that of the GloGEM Positive Degree-Day (PDD) model that they used in their paper." has been removed from the new version of the manuscript according to your suggestion.*

*. About the ranges of the values of anomalies (both x and y-axis), it is not easy to compare the magnitude of sensitivities obtained in our study and found by Bolibar et al. (2022). Indeed, in the paper of Bolibar et al. (2022), the anomalies are calculated and averaged (i) from 660 glaciers covering a wide range of elevations, (ii) from glacier-wide mass blances of these glaciers, (iii) over the 1967-2015 period.*
*As seen in Figure 3, the response of annual mass balance to air temperature and to winter accumulation is different according to the elevation. The magnitude of these sensitivities are different. It is also different from a glacier to another one.*
*The ranges of the anomalies are different but the anomalies covered in our analyses are sufficient to highlight nonlinearities.*

*. Why did we decide to display results for selected elevations (or point mass balances)?*
*The purpose of our study is to show that responses in MB are not linear in response to temperature or precipitation changes even using a simple degree-day model. We used the point mass balances because they are free from dynamics impact (surface changes). The surface changes which influence the glacier-wide mass balance can lead to additional non-linearities. But here, our topic is to discuss the response of surface mass balance using TI model only.*

Fig 4/5: The bottom panels should by labelled "Cumulative daily mass balance (m w.e.)". In my opinion. "Mass balance (m w.e. a-1)" is not correct in this context as a daily time series is shown.

*The changes have been done.*

---

## Referee Report (RR1)

This Brief communication is a revised version. I would like to comment that the track changes in yellow is not really a full track change. One does not see deleted words or edits, just parts where changes are made in marked in yellow. I recommend the authors use a real track change version along with a clean version to make it easier for the authors in new papers or revised versions of this paper.

I find the paper suitable for publishing as a TC Brief communication with its form of three figures and short text. The topic is indeed relevant and timely. The authors have responded to the reviews and even published more material. Could the extra material be published as supplement to the paper? I am not sure if this is allowed in the BC format, but if not it is available through the open discussion.

I recommend some modifications to address the tone in the manuscript as it is clear from the review round that some heat was caused by the tone. I agree with the editor *that the article's reach will be much greater if it was to be framed as a thorough, yet neutral analysis, rather than an open criticism of previous work (in the original submission, some of the wording suggested the latter).'* In my opinion the authors have mended it partly, but could still remove phrases like questioned in the abstract and surprisingly in the main text. You can get your point stated without this by reformulating to a gentler tone. Such experienced researchers should be able to do so. E.g. they write that they have removed the sentence on line 36 "Their paper questions the use of temperature-index models for projections of glacier-mass changes in response to global warming ." But they still have a similar sentence in their abstract, revised vs line 15 ' ..has recently been questioned'
This sentence should be reformulated to be more neutral.

In the following I address others parts where this can be done. I don't think this will be in the way for their message, rather make it clearer and more neutral.

The current lines 34-41 could also be written more neutral, e.g. by removing on line 35, unlike linear statistical and temperature-index models.

And instead of writing on line 37-39
*Bolibar et al. (2022) argue that temperature-index models, widely used to simulate the large-scale evolution of glaciers, provide only linear relationships between positive degree-days (PDDs), solid precipitation and SMB.* Change to

**Bolibar et al. (2022) argue that temperature-index models, widely used to simulate the large-scale evolution of glaciers, can be suitable for steep mountain glaciers, but may be less suitable for some scenarios and flatter glaciers and ice caps due to linear sensitivities in such mass balance models.**

In this way it is more neutral and then can have a natural transition to the point of your study. The authors emphasize the aims many times in the response but it could be more clearly written here. I thus recommend the use of 'only' here to be avoided. It is like using never and always, they are rarely true and easy to argue against.

You write in the response *'Our unique purpose is to show that temperature-index models are able to capture nonlinear responses of glacier mass balance (MB) to high deviations in air temperature and solid precipitation.'* Why not merge this with line on 39 starting Here we… you can make your point clearer. E.g. **'In this paper we perform numerical experiments with a classic and simple temperature-index model. Our unique purpose is to demonstrate that temperature-index models are able to capture nonlinear responses of glacier mass balance (MB) to high deviations in air temperature and solid precipitation.'**

Line 107. …I suggest dropping the last part of the sentence *'contrary to the conclusions of Bolibar et al. (2022) relative to temperature index model.'* Your point will still be valid without it.
L122 . similarly drop *' and this is also inconsistent with the conclusions of Bolibar et al. (2022) relative to the 123 sensitivity of temperature-index models'* your point is still clear even if you drop this part of the

sentence. As you write in your response *"Our unique purpose is to show that temperature-index models are able to capture nonlinear responses of glacier mass balance (MB) to high deviations in air temperature and solid precipitation.'* Focus on this aim.

L135. Consider rewrite the sentence *the opposite results ...are paradoxical* …I suggest at least to drop *'are paradoxical'* you need not have this in the paper to demonstrate your point.

L141-L152. This whole section should be rewritten to make it more neutral. I emphasize it is fair to discuss the choice of or interpretation of models/other studies, but the tone can be adjusted. Words like *'claim', 'even more surprising'* etc seems a bit unneeded. Try to make the points/text more neutral.

Line 150, rev vs. they state that '. Surprisingly, we detect sensitivity to winter accumulation, contrary to the Bolibar et al. (2022) findings using their ANN (Fig. 2 and 3). ' -> Please reformulate and avoid using phrases like Surprisingly to sound more neutral.

L156:
Rewite this '*These results question those of Bolibar et al. (2022), who argue that temperature-index models provide only linear relationships between positive degree-days (PDDs), solid precipitation and SMB.'*
Why not rather write:
**These results highlight that temperature-index models are able to capture nonlinear responses of glacier mass balance (MB) to high deviations in air temperature and solid precipitation.' To emphasize your purpose.**

Delete: *'We tried to understand the cause of this discrepancy.'* Then you can continue with:
**Bolibar et al. (2022) compared the response of SMB to climate forcing (air temperature, winter and summer snow falls).**

---

## Author Response (AR2)

**Reply to Reviewer 4 :**

*We have complied with all the comments made by Reviewer 4 (see the point-by-point response below). The replies are in italics.*

**General comments** :

This Brief communication is a revised version. I would like to comment that the track changes in yellow is not really a full track change. One does not see deleted words or edits, just parts where changes are made in marked in yellow. I recommend the authors use a real track change version along with a clean version to make it easier for the authors in new papers or revised versions of this paper. I find the paper suitable for publishing as a TC Brief communication with its form of three figures and short text. The topic is indeed relevant and timely. The authors have responded to the reviews and even published more material. Could the extra material be published as supplement to the paper? I am not sure if this is allowed in the BC format, but if not it is available through the open discussion. I recommend some modifications to address the tone in the manuscript as it is clear from the review round that some heat was caused by the tone. I agree with the editor 'that the article's reach will be much greater if it was to be framed as a thorough, yet neutral analysis, rather than an open criticism of previous work (in the original submission, some of the wording suggested the latter).' In my opinion the authors have mended it partly, but could still remove phrases like questioned in the abstract and surprisingly in the main text. You can get your point stated without this by reformulating to a gentler tone. Such experienced researchers should be able to do so. E.g. they write that they have removed the sentence on line 36 "Their paper questions the use of temperature-index models for projections of glacier-mass changes in response to global warming ." But they still have a similar sentence in their abstract, revised vs line 15 ' ..has recently been questioned' This sentence should be reformulated to be more neutral.

*Many thanks for your comments. We agree. It is crucial that our paper does not sound as reproach. As mentioned earlier in the round of submission, our unique purpose is to show that temperature-index models are able to capture nonlinear responses of glacier mass balance (MB) to high deviations in air temperature and solid precipitation.*
*In the abstract, « has been questioned » has been removed and the sentence has been reformulated :*
*« The ability of temperature-index models to capture nonlinear responses of glacier surface-mass balance (SMB) to high deviations in air temperature and solid precipitation has recently been discussed in the context of mass-balance simulations employing advanced machine-learning techniques. "*
*We think it is difficult to be more neutral.*
*We also removed "and simple" in the abstract to keep the 100-word upper limit as required in the BC format.*
*"Surprisingly" has been removed (l. 130).*

*About extra material which could be published as supplement to the paper:*

*Our analysis led to find several anomalies in the paper of Bolibar et al. For instance, from our results, we found that the response of SMB to summer snow fall anomalies, using degree day model is almost linear. The annual mass balance anomaly cannot be detected because the summer snow falls are low and do not affect significantly the sensitivity. Although the summer snow fall can affect the summer mass balance, the response to summer mass balance anomaly is almost linear to summer snow fall changes. In addition, in the future, the summer snowfall will be presumably increasingly lower.*
*It is very surprising that Bolibar et al. (2022) encountered that the strongest nonlinear response (from a statistical point of view) came from summer snowfall anomalies. However, this new discrepancy with Bolibar et al. study is beyond the scope of our paper. Again, the topic of our paper is to demonstrate that the responses of degree-day model to temperature and winter accumulation are nonlinear.*

*In fact, it is not very pleasant and comfortable to write a paper which on some points goes against the conclusions of another paper. Given that temperature index models are widely used by the glaciological community for glacier projections in large scale-studies over the 21ˢᵗ century, and according discussions with several scientists involved in this research we contacted about this subject, we believed that it was crucial to clarify these issues. We hope it has been done with our paper.*
*However, we will be very happy when the process will be over.*
*Many thanks again to Reviewer and Editor to help us to make a paper more neutral.*

In the following I address others parts where this can be done. I don't think this will be in the way for their message, rather make it clearer and more neutral. The current lines 34-41 could also be written more neutral, e.g. by removing on line 35, unlike linear statistical and temperature-index models.

*Agree. « unlike linear statistical and temperature-index models » has been removed.*

And instead of writing on line 37-39 Bolibar et al. (2022) argue that temperature-index models, widely used to simulate the large-scale evolution of glaciers, provide only linear relationships between positive degree-days (PDDs), solid precipitation and SMB. Change to Bolibar et al. (2022) argue that temperature-index models, widely used to simulate the large-scale evolution of glaciers, can be suitable for steep mountain glaciers, but may be less suitable for some scenarios and flatter glaciers and ice caps due to linear sensitivities in such mass balance models. In this way it is more neutral and then can have a natural transition to the point of your study.
*Agree. The sentence has been replaced according to your suggestion*

The authors emphasize the aims many times in the response but it could be more clearly written here. I thus recommend the use of 'only' here to be avoided. It is like using never and always, they are rarely true and easy to argue against. You write in the response 'Our unique purpose is to show that temperature-index models are able to capture nonlinear responses of glacier mass balance (MB) to high deviations in air temperature and solid precipitation.' Why not merge this with line on 39 starting Here we... you can make your point clearer. E.g. 'In this paper we perform numerical experiments with a classic and simple temperature-index model. Our unique purpose is to demonstrate that temperature-index models are able to capture nonlinear responses of glacier mass balance (MB) to high deviations in air temperature and solid precipitation.'

*Many thanks. We have adopted your suggestion.*

Line 107. …I suggest dropping the last part of the sentence 'contrary to the conclusions of Bolibar et al. (2022) relative to temperature index model.' Your point will still be valid without it.

*Agree. The last part of the sentence « contrary to the conclusions of Bolibar et al. (2022) relative to temperature index model. » has been removed.*

L122 . similarly drop ' and this is also inconsistent with the conclusions of Bolibar et al. (2022) relative to the sensitivity of temperature-index models' your point is still clear even if you drop this part of the sentence. As you write in your response "Our unique purpose is to show that temperature-index models are able to capture nonlinear responses of glacier mass balance (MB) to high deviations in air temperature and solid precipitation.' Focus on this aim.

*Agree. The part of the sentence "and this is also inconsistent with the conclusions of Bolibar et al. (2022) relative to the sensitivity of temperature-index models" has been removed.*

L135. Consider rewrite the sentence the opposite results …are paradoxical …I suggest at least to drop 'are paradoxical' you need not have this in the paper to demonstrate your point.

*Agree. « are paradoxical » has been removed.*

L141-L152. This whole section should be rewritten to make it more neutral. I emphasize it is fair to discuss the choice of or interpretation of models/other studies, but the tone can be adjusted. Words like 'claim', 'even more surprising' etc seems a bit unneeded. Try to make the points/text more neutral.

*Agree. The sentences of this section have ben reformulated to make the points more neutral.*

Line 150, rev vs. they state that '. Surprisingly, we detect sensitivity to winter accumulation, contrary to the Bolibar et al. (2022) findings using their ANN (Fig. 2 and 3). ' -> Please reformulate and avoid using phrases like Surprisingly to sound more neutral.

*Agree. « Surprising » and « surprisingly » have been removed. The sentences have ben reformulated to make the points more neutral.*

L156: Rewite this 'These results question those of Bolibar et al. (2022), who argue that temperature-index models provide only linear relationships between positive degree-days (PDDs), solid precipitation and SMB.' Why not rather write: These results highlight that temperature-index models are able to capture nonlinear responses of glacier mass balance (MB) to high deviations in air temperature and solid precipitation.' To emphasize your purpose. Delete: 'We tried to understand the cause of this discrepancy.' Then you can continue with: Bolibar et al. (2022) compared the response of SMB to climate forcing (air temperature, winter and summer snow falls).

*Agree. However, given that the first sentence of Conclusions is very similar to your suggestion, we merged the two sentences: « From numerical experiments with a classic and simple temperature-index model, our results highlight that temperature-index models are able to capture nonlinear responses of glacier mass balance (MB) to high deviations in air temperature and solid precipitation, unlike Bolibar et al. (2022) study. »*